

# Downscaling and monitoring the extreme flood events in the Yangtze River Basin based on GRACE/GRACE-FO satellites data

Jingkai Xie[1], Yue-Ping Xu[1], Hongjie Yu[1], Yan Huang[2], Yuxue Guo[1]

[1]Institute of Hydrology and Water Resources, Zhejiang University, Hangzhou, 310058, China
5   [2]Changjiang Water Resources Commission of the Ministry of Water Resources, Wuhan, 43000, China

*Correspondence to*: Yue-Ping Xu (yuepingxu@zju.edu.cn)

**Abstract.** Gravity Recovery and Climate Experiment (GRACE) and its successor GRACE Follow-on (GRACE-FO) satellites provide terrestrial water storage anomaly (TWSA) estimates globally that can be used to monitor the floods in various regions at monthly intervals. However, the coarse temporal resolution of GRACE/GRACE-FO satellites data has 10 been limiting its applications at finer temporal scales. In this study, TWSA estimates have been reconstructed and then temporally downscaled into daily values based on three different learning-based models, namely multi-layer perceptron (MLP) model, long-short term memory (LSTM) model and multiple linear regression (MLR) model. Furthermore, a new index incorporating temporally downscaled TWSA estimates combined with daily average precipitation anomalies is proposed to monitor the severe flood events at sub-monthly time scales for the Yangtze River Basin (YRB), China. The 15 results indicated that (1) the MLP model shows the best performance in reconstructing monthly TWSA with RMSE = 10.9 mm/month and NSE = 0.89 during the validation period; (2) the MLP model can be useful in temporally downscaling monthly TWSA estimates into daily values; (3) the proposed normalized daily flood potential index (NDFPI) facilitates robust and reliable characterization of severe flood events at sub-monthly time scales; (4) the flood events can be monitored by the proposed NDFPI earlier than traditional streamflow observations with respect to the YRB and its individual basins. 20 All these findings can provide new opportunities for applying GRACE/GRACE-FO satellites data to investigations of sub-monthly signals and have important implications for flood hazard prevention and mitigation in the study region.

## 1 Introduction

Extreme floods, as one of the most destructive natural hazards, can result in significant damage to structures and agriculture (Dottori et al., 2018). According to the report published by the United Nations Office for Disaster Risk Reduction (UNDRR), 25 the total economic loss induced by flood is up to \$651 billion (USD) worldwide from 2000 to 2019 (https://www.undrr.org/publication/human-cost-disasters-overview-last-20-years-2000-2019). Meanwhile, floods are projected to become more frequent and extreme under global warming as it can substantially amplify the water holding capacity of the air and increase the occurrence of extreme precipitation events (Slater et al., 2016). Therefore, monitoring



extreme flood events has long been a hot topic for hydrologists and decision makers around the world (Berghuijs et al., 2016;
Smith et al., 2015; Tanoue et al., 2020; Tellman et al., 2021; Thieken et al., 2005).

The Yangtze River Basin (YRB) has been regarded as one of the most sensitive and vulnerable regions that suffered from severe extreme floods due to its highly uneven rainfall pattern (Zhang et al., 2021). During the past decades, the increasingly intensified human activities and climate change have significantly changed the hydrological cycles in the YRB and thus accelerated the variation of flood characteristics in this region. It has been found that both the frequency and severity of
extreme flood events generally showed an upward trend in the YRB in the recent decades owing to substantial changes in climate, infrastructure and land use (Huang et al., 2015; Liu et al., 2019; Yang et al., 2021; Zhang et al., 2008). For example, in summer 2020, the basin has experienced one of the most extreme flood events on record, which ultimately resulted in a great economic and social loss of $27.68 billion (USD) across the entire basin (Jia et al., 2021).

Contrary to traditionally ground-based observations or hydrological models, the launches of Gravity Recovery and Climate
Experiment (GRACE) twin satellites in 2002 and its successor GRACE Follow-on (GRACE-FO) satellites in 2018 can provide a new methodology for retrieving terrestrial water storage anomalies (TWSA) in real time globally by measuring temporal variations in Earth's gravity field (Ahmed et al., 2021; Tapley et al., 2004). TWSA derived from GRACE/GRACE-FO satellites comprises all the surface and subsurface water over land, which can be used to monitor the hydrologic variations in response to extreme weather events. In this case, GRACE/GRACE-FO observations have been widely applied
to assess the potential flood risks for a specific region. For example, Reager et al. (2009) proposed a flood potential index estimated by using the monthly average precipitation anomalies and GRACE-derived TWSA to characterize the potential flood risks from regional to global scales. Xiong et al. (2021) developed a novel integrated flood potential index by linking the flood potential index derived from six GRACE products based on the copula function, which was further used to identify and characterize the floods with different intensities over the study region. A summary of relevant literature on detecting
extreme flood events using GRACE/GRACE-FO data has been listed in Table 1.

Previous studies have clearly indicated that the proposed indices using GRACE/GRACE-FO data can better reflect the evolution of flood events than traditional evaluation indices, such as standardized precipitation index (SPI) and standardized precipitation evapotranspiration index (SPEI), because the GRACE/GRACE-FO observations can measure the vertically integrated water storage over regions (Yan et al., 2021; Yin et al., 2021). However, all these studies mainly focus on
detecting the extreme flood events at monthly intervals while monitoring the flood events and its hydrological impacts at finer temporal scales remains a major challenge due to the coarse temporal resolution (i.e. monthly) of GRACE/GRACE-FO data. To date, very few attentions have been paid to monitor flood events at sub-monthly time scales using GRACE data. Given the rapid occurrence and evolution of some extreme events within a short period, there is a great need to monitor the flood events at a finer temporal resolution (e.g. day) using the temporally downscaled GRACE data, which has important implications for better understanding the mechanisms of extreme flood events development in the YRB. Therefore, we aim



to downscale the TWSA estimates derived from GRACE/GRACE-FO satellites data into daily values and demonstrate its application to monitor the extreme flood events at sub-monthly time scales for the YRB. The temporally downscaled TWSA data could be valuable for understanding the effects of climate change on the hydrological cycle and providing important implications of flood hazard prevention and water resource management over this region.

The rest of this paper is mainly organized as follows. In Section 2, descriptions of the study area are presented. In Section 3 and Section 4, the datasets and methods used in this study are introduced respectively. In Section 5, monthly TWSA estimates obtained from original GRACE/GRACE-FO satellites data are temporally downscaled into individual values at daily time scales based on the methodology proposed in this study. Meanwhile, a new index incorporating temporally downscaled TWSA estimates and daily precipitation is proposed to detect extreme flood events occurred in summer 2020

across the YRB and its individual basins. Then, the discussions about the temporally downscaled GRACE/GRACE-FO satellites data and its capacity to monitor extreme flood events are presented in Section 6. We also explain the reasons why the new proposed index can monitor extreme flood events across the YRB in this section. Finally, we present a summary of this study in Section 7.

**Table 1.**

**2 Study area**

The Yangtze River (also termed as Changjiang River) is the longest river in China with a length of about 6,300 km. It originates from the Tanggula Mountains of Qinghai-Tibetan Plateau and eventually empties into the estuary of the East China Sea after spanning over eleven provinces in China. The YRB (90 °E - 122 °E, 25 °N - 35 °N) has a total drainage area of 1.81 million km2, which accounts for approximately 20% of the total area of the mainland China. The terrain of the YRB

generally decreases from west to east and shows a three-step ladder distribution with altitudes ranging from -142 m to 7143 m above the sea level. The entire YRB consists of three main parts, that is, the Upper (upstream region above the Yichang station), the Middle (region between the Yichang station and the Hukou station) and the Lower (downstream region below the Hukou station) subbasins (shown in Fig. 1).

The YRB is located in typically subtropical and temperature climate zones, which is dominated by three types of monsoons,

namely the Siberian northwest monsoon winds in winter, the Indian southeasterly monsoon winds and the East Asian monsoon in summer (Kong et al., 2020). According to the observations from meteorological stations, the mean annual air temperature of this basin ranges from 14.4 ℃ to 15.4 ℃ and mean annual precipitation ranges from 1049 mm to 1424 mm during the study period. Under the joint effects of monsoon activities and seasonal motions of subtropical highs, more than 85% of the annual precipitation occurs in the wet season from April to October, which further increases the risks of extreme

floods in the middle and lower reaches of the Yangtze River (Huang et al., 2015; Yang et al., 2010). Additionally, by the end





of 21st century, projections show a significant upward trend of the annual precipitation over the YRB according to the latest study (Yue et al., 2021).

The YRB is one of the most important regions in China because it accommodates approximately 33% of China's total population (Huang et al., 2021), accounts for over 36% China's total water resources, and contributes more than 46% of

China's total Gross Domestic Product (GDP) according to the statistics collected by Yangtze River Conservancy Commission of Ministry of Water Resources. The YRB not only sustains many hydro-electrical industries, such as the Three Gorges Corporation, but also provides freshwater resources for neighboring regions to alleviate the pressure of water scarcity through the South-to-North Water Diversion Project (Long et al., 2020; Zhang et al., 2021). Furthermore, the YRB can play a critical role in flood control, crop irrigation, power generation, and ecological conservation (Chao et al., 2021; Wang et al.,

2020a). More information about the location and topography of the YRB can refer to Fig. 1.

**Figure 1.**

## 3 Data

### 3.1 Terrestrial water storage derived from GRACE/GRACE-FO satellites data

GRACE and GRACE-FO data can provide global TWSA at monthly scales. In this study, the average of three types of

GRACE and GRACE-FO solutions is estimated in order to characterize the variations of TWSA in the YRB and its individual basins during the period of 2003-2020, all of which are the latest versions of Release Number 06 (RL06). These products are provided by the Center for Space Research (CSR, at the University of Texas at Austin) (Save et al., 2016), the Goddard Space Flight Center (GSFC, at NASA) (Loomis et al., 2019) and the Jet Propulsion Laboratory (JPL, at NASA and California Institute of Technology, California) (Landerer et al., 2020) respectively. All these GRACE and GRACE-FO

solutions represented by equivalent water thickness units (mm) are anomalies relative to the time-mean baseline during January 2004 - December 2009. It should also be noteworthy that GRACE data in a few months are not available because of the problem of "battery management". In addition, there existed a gap period for 11 consecutive months from July 2017 to May 2018 between the GRACE and GRACE-FO satellites. Here we have not filled the data gaps between the two GRACE satellites with linear interpolation since it may not fully describe the seasonal variation of TWSA during these missing

months. All these GRACE and GRACE-FO satellites data are available at the website of https://podaac.jpl.nasa.gov.

### 3.2 Meteorological data

In this study, daily time series of precipitation and temperature from 2003-2020 are provided by the China Meteorological Administration (CMA) (http://data.cma.cn/) with a total of 150 National Meteorological Observatory stations distributed in the YRB (shown in Fig. 1). Areal precipitation in the YRB and its individual basins at daily scales can be calculated





according to the Thiessen polygon method. Monthly precipitation for regions is calculated by summing all daily values of precipitation. Meanwhile, areal temperature in the YRB and its basins at daily time scales are calculated by directly averaging the respective daily temperature from all meteorological stations over regions. Similarly, monthly temperature estimates are calculated by summing all daily values of temperature.

### 3.3 In-situ streamflow data

From the Yangtze River Conservancy Commission of Ministry of Water Resources, daily streamflow observations during the period of 2013-2020 can be obtained at the Shigu hydrological station, the Yichang hydrological station, the Hukou hydrological station and the Datong hydrological station (shown in Fig. 1). More specifically, the Shigu station represents the outlet of the Source regions of the Yangtze River Basin (SYRB), the Yichang station represents the outlet of the Upper regions of the Yangtze River Basin (UYRB), the Hankou station represents the outlet of the Upper and the Middle regions of

the Yangtze River Basin (UMYRB), and the Datong station represents the outlet of the entire Yangtze River Basin (YRB). Meanwhile, extreme flood events in the YRB and its individual basins during the study period can be extracted from daily time series of streamflow observed from the above hydrological stations.

### 3.4 Soil moisture storage

As documented in Xie et al. (2019a), soil moisture storage (SMS), as one of critical components of terrestrial water storage,

usually shows a significantly positive correlation with variations of regional TWSA. Therefore, in this study we adopt the SMS (kg/m2) with a spatial resolution of $0.25° \times 0.25°$ from the Global Land Data Assimilation System version 2.1 (GLDAS 2.1) Noah land surface model to estimate their correlations with regional TWSA derived from the GRACE and GRACE-FO satellites data. This product can provide the simulations of SMS at four different depths of soil layers from 0 to 200 cm, that is, 0 - 10 cm, 10 - 40 cm, 40 - 100 cm and 100 - 200 cm depths per three hours. To keep consistent with TWSA,

the original value of SMS should be transferred into soil moisture storage anomaly values (SMSA) after subtracting the time-mean baseline during the period of 2004-2009. Original SMS derived from GLDAS 2.1 Noah land surface model can be aggregated o daily and monthly estimates as follows:

$$SMSA_{daily} = SMS \times 8 - SMS_{baseline}, \tag{1}$$

$$SMSA_{monthly} = SMS \times 8 \times N - SMS_{baseline}, \tag{2}$$

where $SMSA_{daily}$ (mm) and $SMSA_{monthly}$ (mm) represent daily SMSA and monthly SMSA respectively; $SMS$ represents the original SMS estimates derived from GLDAS 2.1 Noah land surface model; $SMS_{baseline}$ represents the baseline average during 2004-2009; $N$ represents the number of days in a specific month. More specific information about the datasets used in this study can be found in Table 2.



**Table 2.**

## 150    **4 Methods**

To better monitor the extreme flood events occurred in the YRB, monthly TWSA obtained from original GRACE/GRACE-FO satellites data are temporally downscaled into individual values at daily time scales based on the methodology proposed in this study. A detailed flow diagram of our study is given in Fig. 2, which is made of four steps. In Step 1, meteorological observations including precipitation, temperature provided by CMA and SMSA derived from the GLDAS 2.1 Noah land

surface model are jointly used as model inputs to establish the relationship with detrended GRACE/GRACE-FO satellites data. In Step 2, the relationship between TWSA estimates and all hydro-climatic factors at monthly time scales for the YRB can be built by using three different learning-based models, namely MLP model, LSTM model and MLR model respectively. After comparing the performances of each model in simulating monthly TWSA estimates under all three scenarios, the scaling properties of the model (i.e. calibrated model parameter sets) that shows the best performance in simulating monthly

TWSA estimates are identified and retained. In Step 3, daily time series of meteorological observations and the SMSA from the GLDAS 2.1 Noah land surface model are reselected as model inputs of the relationship established in Step 3 assuming that scaling properties at the monthly time scales are valid at the daily time scales. And hence daily TWSA estimates can be temporally downscaled from monthly TWSA estimates by using the calibrated model parameter sets that have been identified and retained in Step 3. In Step 4, daily time series of TWSA are further applied to monitor the flood events at sub-

monthly time scales for different basins in the YRB according to the new proposed index.

**Figure 2.**

Specifically, three types of models, namely, the artificial neural network (ANN), the recurrent neural network (RNN) and the multiple linear regression (MLR) are served as the statistical downscaling methods. In order to keep a fair comparison, we will choose identical inputs and outputs in the process of training these three models. Furthermore, the GRACE satellites can

provide TWSA estimates under the joint effects of human activities and climatic variability (Xie et al., 2019b). As pointed out by previous studies (Humphrey and Gudmundsson, 2019; Khorrami et al., 2021; Shah et al., 2021), many long-term trends in GRACE data are primarily caused by frequent human activities such as persistent groundwater overexploitation and massive construction of large reservoirs. For example, the YRB is a typical region strongly influenced by various human activities, such as the construction of Three Gorges Reservoir and intense human water consumption (Huang et al., 2015;

Yao et al., 2021). In this study, the linear trends have been removed from the original time series of TWSA in the training and calibration periods because hydro-climatic factors may not fully simulate these long-term trends, all of which are mainly arising from human activities, such as the water withdrawals and reservoir operation over the study region (Rodell et al., 2018). More detailed descriptions about the methods used in this study are given as follows.





### 4.1 Multi-layer perceptron neural network (MLP)

The ANN is a black box model which has the ability to imitate the thought processes of the human brain and thus can be applied to deal with complex and nonlinear problems (Bomers et al., 2019; Boucher et al., 2020; Lecun et al., 2015; Wang et al., 2020b). Among different types of ANNs, the multi-layer perceptron neural network (MLP) with the Levenberg-Marquardt Back Propagation training algorithm is the most widely used method as it requires relatively less time in the process of convergence (Rumelhart et al., 1986; Xie et al., 2019a). Therefore, a three-layer MLP model and the Logarithmic

Sigmoid as a transfer function are jointly used for temporal downscaling in this study, which has been proved to be effective and reliable in statistical downscaling (Nourani et al., 2018; Sharifi et al., 2019). This MLP model consists of three parts, namely, an input layer, a hidden layer and an output layer, all of which finally form a network through many neurons. Meanwhile, the weights, which are connections between different neurons, can adjust as learning proceeds until the most optimum network is derived in this process (Fig. 3(a)).

In this study, the variables included in the input layer are precipitation, temperature and SMS whereas the variable included in the output layer is detrended TWSA. Based on trial-by-error, the most optimal number of hidden neurons is set to five. After minimizing the discrepancy between the simulated TWSA with the observed results at the output layer, the most optimal network architecture can be finally obtained.

### 4.2 Long short-term memory network (LSTM)

The recurrent neural network (RNN) (Rumelhart et al., 1986) is a unique type of deep learning algorithm that was developed to process sequential data and predict future trends. One of the most dominated features of the RNN layer is a unique feedback connection which can allow past information to continuously affect the current output. The characteristics of all related time series data can be eventually learned through this structure. Long short-term memory network (LSTM) is one of the most representative RNNs as it has the fabulous memory ability and can effectively avoid the vanishing gradient problem

existed in other RNNs (Hochreiter, 1997; Guo et al., 2021). Considering the time series characteristics of meteorological data and TWSA data, the LSTM model can be very suitable as a statistical downscaling model for its excellent capacity to process sequence-to-sequence learning problems.

One typical LSTM model usually consists of three layers, that is, an input layer, a hidden layer and an output layer (Fig. 3(b)). Different from other traditional ANNs, the LSTM model replaces the hidden block in RNNs with a memory cell state

coupled with three logic gates, that is, the forget gate, the input gate and the output gate. In the training process, the memory cell state mainly stores the accumulation of past information. The input gate determines how much information of a new input flows into the memory cell state at the current time. Then, the useless information in long-term memory would be forgotten by the forget gate, which determines how much of the former moment is retained to the current time. Finally, the



output gate determines how much information of the memory cell state is used to compute output (Bai et al., 2021; Wu et al.,

2020; Vu et al., 2021). The main formulations of the LSTM model are therefore described as follows:

input gate ($i_t$):

$$i_t = \sigma(W_i x_t + U_i h_{t-1} + b_i), \tag{3}$$

forget gate ($f_t$):

$$f_t = \sigma(W_f x_t + U_f h_{t-1} + b_f), \tag{4}$$

output gate ($o_t$):

$$o_t = \sigma(W_o x_t + U_o h_{t-1} + b_o), \tag{5}$$

potential cell gate ($\tilde{c}_t$):

$$\tilde{c}_t = tanh(W_{\tilde{c}} x_t + U_{\tilde{c}} h_{t-1} + b_{\tilde{c}}), \tag{6}$$

cell gate ($c_t$):

$$c_t = f_t \odot c_{t-1} + i_t \odot \tilde{c}_t, \tag{7}$$

hidden gate ($h_t$):

$$h_t = o_t \odot tanh(c_t), \tag{8}$$

where $i_t$, $o_t$, $c_t$, $\tilde{c}_t$ and $h_t$ represent the input gate, output gate, cell gate, potential cell gate and hidden gate at the time $t$, respectively; $x_t$ represents the standardized input variable at the current time $t$; $h_{t-1}$ represents the hidden state at the time $t$-1;

$c_{t-1}$ represents the previous cell state that provides the past information at the time $t$-1; $tanh$ represents the hyperbolic tangent; $\sigma$ denotes the logistic sigmoid function that is usually served as the gate activation function; $\odot$ denotes the element-wise multiplication of vectors; $W$, $U$ and $b$ represent the input weights, the recurrent weights and the biases of each gate, respectively, all of which are usually estimated during the learning process in matching the training data according to the adaptive moment estimation (ADAM) optimizer.

**Figure 3.**

### 4.3 Multiple linear regression (MLR)

The multiple linear regression (MLR) is a typical statistical approach that can be applied to establish the relationships between inputs and outputs (Sousa et al., 2007). This approach has a wide range of hydrological applications since it can well explain the linkage between various variables (Lyu et al., 2021; Ramesh et al., 2020; Sun et al., 2020). Here we assume



that the GRACE/GRACE-FO derived TWSA is linearly regressed onto the meteorological variables (i.e. precipitation and temperature) and the SMS obtained from GLDAS 2.1 simultaneously, that is:

$$y = \sum_{i=1}^{3} a_i \times x_i + b, \tag{9}$$

where $y$ represents TWSA at monthly (or daily) scales; $x_i$ ($i = 3$) represents three independent inputs including precipitation, temperature and SMS at monthly (or daily) scales; $a_i$ represents the corresponding regression coefficients of each input,

which can be calculated by the least-squares regression method; $b$ represents a constant offset. Similar to LSTM and MLP, the MLR model is also applied at regional scales to better show the temporal variation of downscaled TWSA during the extreme flood events occurred in the YRB.

**4.4 Flood event selection**

A nonparametric algorithm is adopted to identify runoff events in this study (Tarasova et al., 2018). A brief procedure of this

algorithm is described as follows: (1) picking out local minima within nonoverlapping five-day windows with respect to the entire streamflow time series; (2) examining the extracted series of minima with the goal of finding turning points, all of which are usually defined as the points that are at least 1.11 times smaller than their neighboring minima; (3) reconstructing the base flow hydrograph according to the linear interpolation between the turning points, which are previously obtained in Step (2); (4) screening the streamflow time series to identify runoff events after the separation of base flow. Traditionally, a

typical runoff event can be characterized by three main components, namely peak, beginning, and end points. A peak refers to the maximum of streamflow for a specific period. The beginning point refers to the closest point in time when total runoff is equal to base flow before the peak. The end point denotes the point in time when the total runoff as soon as has fallen to the base flow level from peak.

**4.5 Daily flood potential index**

The flood potential index provides a surrogate measure of the potential flood risks for a specific region, which can be obtained from monthly average precipitation anomalies and GRACE-derived TWSA (Reager et al., 2009). In this study, we further propose a new normalized daily flood potential index (NDFPI) with reference to Reager et al. (2009) and Abhishek et al. (2021). Compared to the original flood potential index, the NDFPI can not only provide useful information on the early signs of the region's transition from normal state to a flood-prone situation but also effectively detect the flood events at sub-

monthly time scales, which is calculated via the following steps:

$$TWSA_{def}(t) = TWSA_{max} - TWSA(t-1), \tag{10}$$





where $TWSA_{def}(t)$ (mm) represents the terrestrial water storage deficit for a specific day ($t$) that is defined as the difference between the historic storage anomaly time series maximum during the entire period ($TWSA_{max}$) and the storage amount from the previous day ($TWSA(t-1)$).

Then, the daily flood potential amount (DFPA) is further calculated as follows:

$$DFPA(t) = P(t) - TWSA_{def}(t) = P(t) - (TWSA_{max} - TWSA(t-1)), \qquad (11)$$

where $DFPA(t)$ (mm) represents the daily flood potential amount for a specific day ($t$); $P(t)$ (mm) represents the daily precipitation; $TWSA(t-1)$ (mm) represents the TWSA from the previous day ($t-1$).

Finally, we can calculate the normalized daily flood potential index (NDFPI) from the DFPA with the goal of removing the

effects of hydrological heterogeneity varying from region to region and the typical difference between the storage change and precipitation that may not always result in floods (Reager et al., 2009), which can be described as follows:

$$NDFPI(t) = \frac{DFPA(t) - DFPA_{min}}{DFPA_{max} - DFPA_{min}}, \qquad (12)$$

where $DFPA_{max}$ and $DFPA_{min}$ represent the maximum DFPA and minimum DFPA during the study period respectively. The NDFPI indicates the corresponding probability of flood occurrence with a range from 0 to 1. More flood is likely to

occur when the NDFPI is closer to 1 for a specific region.

**4.6 Model test design**

Monthly TWSA estimates can be reconstructed based on the above three different learning-based models, namely the MLP model, the LSTM model and the MLR model. According to previous findings in Liu et al. (2021), different periods of data used for training (i.e. identification of model parameter sets) and validation can eventually influence the corresponding

performances of a specific model when simulating TWSA. Therefore, we generally design a total of three scenarios according to the way of dividing training periods and validation periods for a specific model. As shown in Fig. 2, periods of GRACE data used for training and validation in each experiment are listed, which include (1) Scenario 1: training period (2003/01-2014/07, a total of 129 months) and validation period (2014/08-2020/12, a total of 56 months), (2) Scenario 2: training period (2005/06-2018/06, a total of 129 months) and validation period (2003/01-2005/05 and 2018/07-2020/12, a

total of 56 months), and (3) Scenario 3: training period (2007/10-2020/12, a total of 129 months) and validation period (2003/01-2007/09, a total of 56 months).

Furthermore, three kinds of statistical measures including root mean square error (RMSE), correlation coefficient (r), and Nash-Sutcliffe efficiency coefficient (NSE) are used in this study as they can jointly measure the matching quality in terms of both magnitude and phase between the simulated and the observed time series. These statistical measures are defined as:





$$RMSE = \sqrt{\frac{\sum_{i=1}^{N}(x_i - x_{oi})^2}{N}}, \tag{13}$$

$$r = \frac{\sum_{i=1}^{N}(x_i - \overline{x_i})(x_{oi} - \overline{x_{oi}})}{\sqrt{\sum_{i=1}^{N}(x_i - \overline{x_i})^2 \times \sum_{i=1}^{N}(x_{oi} - \overline{x_{oi}})^2}}, \tag{14}$$

$$NSE = 1 - \frac{\sum_{i=1}^{N}(x_i - x_{oi})^2}{\sum_{i=1}^{N}(x_{oi} - \overline{x_{oi}})^2}, \tag{15}$$

where $x_i$ and $x_{oi}$ represent simulated and observed values, respectively; $\overline{x_i}$ and $\overline{x_{oi}}$ represent the average of simulated and observed values; $N$ is number of validation values.

## 5 Results

### 5.1 Temporal variation of precipitation, temperature, SMSA, TWSA and streamflow across the YRB during 2003-2020

Fig. 4 shows the monthly time series of SMSA, TWSA, streamflow and the main climatic variables including precipitation and temperature across the YRB during 2003-2020.The results show that monthly TWSA over the YRB has a wide range from -58.0 mm to 130.1 mm during the study period. Monthly TWSA estimated by three GRACE/GRACE-FO solutions changes synchronously with precipitation across the entire YRB, showing a significantly positive correlation between TWSA and precipitation ($r = 0.54$; $p < 0.01$) during the study period. For summer 2020 as an example, a noticeable increase in precipitation has been observed in the YRB during the summer season (Jun to August) in 2020. According to the statistics collected by Yangtze River Conservancy Commission of Ministry of Water Resources, the accumulative rainfall across the entire YRB exceeds 680 mm in summer 2020, which is far more than the mean rainfall (approximately 540 mm) during the same period from 2003 to 2019. Accordingly, TWSA reaches its maximum in July 2020 with an estimate of 130.9 mm during 2003-2020, reflecting the evolution of TWSA in response to heavy rainfall during this period. In addition to precipitation, TWSA is also highly consistent with temperature over the YRB during 2003-2020, showing a positive correlation coefficient of $r = 0.57$ ($p < 0.01$) with monthly temperature.

The GLDAS Noah-derived SMSA and GRACE/GRACE-FO derived TWSA both show a seasonal variation through the entire study period in the YRB, but there exists a significant difference in the intensity of anomalies between them especially in the summer season as depicted in Fig. 4. This phenomenon can be explained by the discrepancies resulted from different components of SMSA and TWSA. Although the SMSA is a decisive component of TWSA for many regions, the latter usually contains some other components, such as the anomalies of surface water and groundwater etc., besides the SMSA (Xie et al., 2021). There exists a significant correlation between TWSA and SMSA with a positive correlation coefficient of $r = 0.84$ ($p < 0.01$), both of which reach to maximum and minimum values almost simultaneously. In general, TWSA shows



a significant correlation with precipitation, temperature and SMSA during the study period, all of which have been therefore selected as the inputs applied to simulate monthly TWSA over different regions.

**Figure 4.**

**5.2 Reconstruction of TWSA by different models**

To achieve the temporal downscaling of monthly TWSA data, we should firstly build the relationships between GRACE/GRACE-FO derived TWSA and various hydro-climatic factors including precipitation, temperature and SMSA at monthly time scales. The results of TWSA are estimated by the mean value in different regions upstream of the corresponding hydrological stations shown in Fig. 1. In this study, three different models including MLP, LSTM and MLR

are adopted to reconstruct TWSA for regions. Table 3 shows the summary of model performances in reconstructing monthly TWSA across the YRB during the study period. GRACE/GRACE-FO satellites data used for training (i.e. identification of model parameter sets) and validation shown in each scenario mainly depend on the periods of series of data as suggested by Liu et al. (2021). According to Table 3, we find that all models including the MLP, the LSTM and the MLR with Scenario 3 show the best performances in simulating monthly TWSA under all three designed scenarios. This result indicates that the

models with Scenario 3 is relatively superior to the models with the other two scenarios when simulating TWSA because the data in Scenario 3 contains more extremely high (or low) values during the study period in the process of training models. Therefore, in the following sections, we decide to directly divide the training periods and validation periods of all these models according to Scenario 3 (shown in Table 3) when simulating monthly TWSA for other regions besides the YRB.

**Table 3.**

Fig. 5 show the comparison between monthly TWSA derived from GRCACE/GRACE-FO satellites data and that simulated by different models for all regions during 2003-2020. The corresponding evaluation values are also presented in this figure. We find that the maximum NSEs between the GRACE/GRACE-FO-derived TWSA estimates and that simulated by models are 0.68, (Fig. 5(a)), 0.82 (Fig. 5(f)), 0.86 (Fig. 5(g)) and 0.89 (Fig. 5(j)) during the validation periods for the SYRB, the UYRB, the UMYRB and the YRB, respectively. The corresponding RMSEs are 13.2 mm/month, 13.7 mm/month, 12.4

mm/month and 10.9 mm/month (validation periods, hereafter) for the SYRB, the UYRB, the UMYRB and the YRB, respectively. In general, the detrended TWSA estimates present consistent values between the observations and the modeled results from 2003-2020 for most regions except for the SYRB, as shown in Fig. 5. Compared to the other regions, all models show a relatively poor performance in simulating monthly TWSA for the SYRB with NSEs less than 0.70 during the validation periods, which can be mainly attributed to the increased uncertainties in precipitation and temperature induced by

the sparse distribution of meteorological stations over this region (shown in Fig. 1).

We further compare separately the performances of all models in simulating monthly TWSA for a specific region. Taking the entire YRB as an example (Fig. 5(j-l)), GRACE/GRACE-FO derived TWSA estimates shows a RMSE of 10.9





mm/month for the MLP-derived TWSA estimates, which is lower than that of 15.1 mm/month for the LSTM-derived TWSA estimates (~39% decrease) and that of 13.3 mm/month for the MLR-derived TWSA estimates (~22% decrease). Meanwhile,

the NSE shows similar improvements when applying the MLP model to simulate TWSA for the YRB (Fig. 5(j-l)), which also can be found in the SYRB (Fig. 5(a-c)) and the UMYRB (Fig. 5(g-i)). In general, the MLP and MLR models achieve high metrics (0.81/12.8 mm/month and 0.75/14.2 mm/month of NSE/RMSE on average for all regions) during the validation periods, both of which are significantly higher than the metrics between the GRACE/GRACE-FO derived TWSA estimates and that simulated by the LSTM model (0.75/14.7mm of NSE/RMSE on average for all regions). For the UYRB (Fig. 5(d-f)),

the MLP model shows a slightly poor performance in simulating TWSA in terms of a higher RMSE (14.7 mm/month) than the LSTM model (14.5 mm/month; ~1.2% increase) and the MLR model (13.7 mm/month; ~7.2% increase). In addition, it seems that the larger the study region, the higher the correspondence between the GRACE/GRACE-FO derived TWSA estimates and that simulated by models for the MLP model. This result can be explained that the large area for a specific region may smooth more uncertainties in GRACE signals and meteorological observations (Long et al., 2015).

Overall, Fig. 5 clearly suggests the MLP model's superior performances in simulating TWSA with an average value of NSE higher than 0.81 and an average value of RMSE lower than 12.8 mm/month during the validation periods for all regions, showing the outstanding capability of MLP model in learning the complicated relationships between TWSA and hydro-climatic factors. As documented in Shu et al. (2007), the MLP model can show its unique superiority and great advantages compared with other statistical models particularly when explaining the underlying processes that have complex nonlinear

interrelationships. The results shown in Fig. 5 also indicate that the MLP model can show a relatively better performance in simulating monthly TWSA than the LSTM model in this study. As described in Zhang et al. (2018), one of main drawbacks of the LSTM model is its complexity compared with the MLP model, which indicates that the LSTM model may not show better performances in simulating time series data than other traditional ANNs models in some cases especially when limited trained data are available. Therefore, in the following discussion, only the MLP model is applied to further achieve the

temporal downscaling of monthly TWSA data for regions.

**Figure 5.**

## 5.3 Temporal downscaling of GRACE/GRACE-FO satellites data

Relationships between monthly TWSA and hydro-climatic inputs with respect to the entire YRB have been fully established as presented in Section 5.2. As documented in Herath et al. (2016) and Requena et al. (2021), the same scaling properties

have been commonly assumed for baseline and future periods in temporal downscaling. Therefore, it is reasonable and acceptable to assume that scaling properties at the monthly time scales are valid at the daily time scales in this study (Kumar et al., 2012). That is, the relationship between temporally downscaled TWSA and daily hydro-climatic inputs is consistent with that previously established by the downscaling model (e.g. the MLP model) at monthly time scales for a specific region.



By merging the daily hydro-climatic inputs into the previously established relationships between TWSA estimates and
hydro-climatic factors based on the MLP model, we can downscale the TWSA estimates from monthly time series to daily
time series for all regions.

Fig. 6 shows daily time series of TWSA temporally downscaled by the MLP model for different regions during 2003-2020.
It can be seen that daily TWSA shows sub-monthly signals in response to changes in hydro-climatic factors as expected.
Both GRACE/GRACE-FO derived TWSA estimates and daily TWSA estimates temporally downscaled by the MLP model
show obvious seasonal cycles and reach to their respective extreme values almost simultaneously. More specific, amplitudes
of daily TWSA estimates are slightly higher (or lower) than monthly TWSA estimates in summer (or winter) seasons from
2003 to 2020. This can be deemed as reasonable because monthly TWSA estimates are defined as the mean average of daily
TWSA estimates for a specific month. It should also be noted that there still exist some discrepancies between temporally
downscaled TWSA at sub-monthly time scales and monthly TWSA estimates derived from GRACE/GRACE-FO satellites
data for the SYRB particularly in some extreme low values, which can be attributed to the relatively poor relationship
between TWSA estimates and hydro-climatic factors for this region as described in Fig. 5(a).

**Figure 6.**

**5.4 Relation between daily TWSA and streamflow during flood events**

Fig. 7 and Fig. 8 show the daily TWSA temporally downscaled by the MLP model and observed streamflow within the YRB
in 2010 and 2020 when extreme flood events occurred according to the information published by the Yangtze River
Conservancy Commission of Ministry of Water Resources. As described in Fig. 7 and Fig. 8, the nonparametric simple
smoothing method introduced in Section 4.4 can effectively identify the corresponding flood events occurred in each region
based solely on the analysis of streamflow time series. It shows an apparent increase in streamflow from the beginning to
peak of all flood events. Accordingly, the daily TWSA shows a distinct increase similar with streamflow during the same
periods as expected. It is also interesting to note that the beginning of increase shown in daily TWSA is earlier than that of
streamflow. This is partly because high antecedent soil moisture, which is the dominant component of TWSA, has been
identified as an important driver of flood events for regions (Reager et al., 2014; Wasko et al., 2019). Meanwhile, this result
indicates that daily TWSA can be potentially useful in building early flood warning systems since it may identify the
extreme flood events much more earlier than streamflow.

**Figure 7.**

**Figure 8.**





**5.5 Monitoring severe flood events based on the proposed NDFPI in summer 2020**

To better detect the extreme events during the wet season, we propose a new index, i.e. NDFPI, by jointly using the temporally downscaled TWSA data and daily precipitation data observed by meteorological stations as introduced in Section

4.5. According to the Yangtze River Conservancy Commission of Ministry of Water Resources, the YRB has suffered from catastrophic flooding in summer 2020 and a total of 33 rivers in the YRB exceeded their historical maximum water levels during this period. Therefore, in this study, the severe flood events occurred in summer 2020 for the YRB will be served as an example to present the capability of NDFPI in detecting extreme flood events. The threshold values of daily streamflow and NDFPI for the 90th percentile floods during 2003-2020 are presented in Fig. 9. According to the results shown in Fig. 9,

the larger threshold values of NDFPI usually indicate severity of flood occurrences increases for a specific region. In addition, the shape of percentile duration curve of daily streamflow across the UYRB (Fig. 9(b)) is different with that shown in other regions. It is noted that the outlet of the UYRB, the Yichang hydrological station, is located approximately 45 km downstream of the Three Gorges Reservoir (shown in Fig. 1), which is one of the largest hydroelectric reservoirs in the world. Given that the operations of the Three Gorges Reservoir can directly affect the streamflow at Yichang station (Yang

et al., 2022), the result shown in Fig. 9(b) is reasonable.

**Figure 9.**

Fig. 10 shows the comparison between basin averaged NDFPI and daily streamflow observations for the 90th percentile floods in summer 2020. The results indicate that the ups and downs of the streamflow observed at different hydrological stations are highly consistent with the NDPFI results through the whole season. For example, the observations of streamflow

from the Shigu hydrological station (Fig. 10(a)) reached its 90th percentile in July 12. In comparison, the NDFPI estimated by temporally downscaled TWSA and daily precipitation reached its 90th percentile in July 4 (Fig. 10 (a)), which is 9 days earlier than that of daily streamflow. As expected, these high streamflow observations during the wet season are usually accompanied by high NDPFI values, which could be attributed to the effects of high precipitation on streamflow during this period. For the YRB (Fig. 10 (d)), daily streamflow detected at the Datong hydrological station reached its 90th percentile in

June 29 and eventually peaked in July 13 with a maximum value of $7.2 \times 10^9$ m$^3$/day, which is in line with the findings in Jia et al. (2021). Accordingly, the series of NDFPI reached its 90th percentile in June 18 with a value of 0.58. In general, Fig. 10 clearly suggests that the proposed NDFPI calculated by temporally downscaled TWSA data and daily precipitation changes synchronously with the reality of flood disasters in summer 2020 for the YRB. Meanwhile, it also indicates that such flood events can be monitored by the proposed NDFPI earlier than traditional streamflow observations.

**Figure 10.**

Previous studies usually focus on monitoring the long-term flood events while the flood events at sub-monthly time scales using GRACE/GRACE-FO satellites data have been limitedly investigated due to the limitation of its temporal resolution (i.e. month) (Gouweleeuw et al., 2018; Long et al., 2014). In this study, however, Fig. 10 clearly shows the incremental





process of TWSA during the wet season using the new proposed NDFPI estimated by temporally downscaled
GRACE/GRACE-FO satellites data and daily precipitation for different regions. This means that the proposed NDFPI has
the great potential to detect the evolution of extreme flood events within the short period. It is also interesting to note that the
NDFPI reached the threshold of different classes of flood events earlier than that defined by streamflow observations during
the wet season in 2020, which can be repeatedly found in the SYRB, the UYRB, the UMYRB and the YRB (Fig. 10)
respectively. The comparison results indicate that the lag time between the threshold values of flood events monitored by the
NDFPI and that monitored by daily streamflow during the wet season ranges from 8 to 15 days for the 90th percentile floods
among all regions in summer 2020, all of which are far less than the temporal resolution of original GRACE/GRACE-FO
satellites data (i.e. month). In addition to the 90th percentile floods, we also compare the basin averaged NDFPI and daily
streamflow observations for the 95th and 99th percentile floods in summer 2020 (shown in Supplement Figure S1-S4). The
results also show that the series of NDFPI reached the threshold values earlier than that of daily streamflow observations for
the 95th and 99th percentile floods. For example, there exists a 11-day lag time between the threshold value of NDFPI and
that of the streamflow observed by Datong hydrological station for the 99th percentile floods in summer 2020 (Fig. S4(d)),
which provides useful information for accurate and timely flood forecasts and can be very beneficial for protecting people
and infrastructure over regions in a changing climate.

## 6 Discussions

**6.1 Extreme flood events monitored by NDFPI**

The comparison results indicate that the proposed NDFPI reached the threshold values of different classes of flood events
earlier than that defined by streamflow observations in summer 2020 with respect to the YRB and its individual basins
(shown in Figure 8 and Figure S1-S4). This is consistent with the results found at the Missouri River basin by Reager et al.
(2014) who indicated that regional TWSA may lead river discharge slightly before the flood season, creating a simple
hysteresis effect between these two time series. It is this effect that provides useful information on the signal of high
streamflow in the coming flood season and the predisposition for flooding over the study region. However, the study of
Reager et al (2014) only demonstrated the application of GRACE data to characterize regional flood potential at monthly
time scales. More accurate information about the complete hydrologic state of a specific region at sub-monthly time scales
during the wet season has been limitedly investigated, which is very vital for flood warnings. Given this in mind, we
proposed a new index, i.e. NDFPI, by jointly using the temporally downscaled TWSA data and daily precipitation data to
better analyze the hydrologic state of the study region during the wet season at finer time scales.

The comparison analysis of the NDFPI and daily streamflow with respect to the YRB may explain the possible reasons why
the NDFPI can detect extreme flood events for a specific river basin. Intense rainfall of long duration can cause continuous





increases in the surface water (e.g. water stored in lakes and wetlands), soil moisture storage and groundwater storage that

are totally represented by TWSA in this study through the process of infiltration. Many studies also revealed that changes in surface water, soil moisture and groundwater under intense rainfall can exert obvious effects on the status of regional TWSA (Döll et al., 2012; Felfelani et al., 2017; Sinha et al., 2019; Velicogna et al., 2012). All these changes may ultimately result in the saturation of aquifer over regions. However, the saturated state of aquifers is not persistent because there is a great need for the basin to relieve its saturated state by discharging excessive water stored on and below the land surface into the river

channels, which may eventually lead to the dramatic increase in streamflow and greatly increase the risk of widespread and damaging regional flooding.

**6.2 Advantages of detecting extreme flood events based on temporally downscaled TWSA**

The traditional flood monitoring approaches mainly provide useful information about the evolution of flood events over the study region through the measurements of rainfall and streamflow. All these measurements largely depend on the in-situ

hydrological stations and rainfall gauging stations distributed over the regions, which are difficult to achieve in some regions with harsh environment and climatic conditions. In comparison, satellite remote sensing has no such limitation of traditional point-based observations, making it a promising approach to monitor extreme flood events particularly in some poorly gauged basins. Given the large spatial extent, complicated climatic condition, and inaccessible hydrological observations for some high-altitude regions (e.g. SYRB), GRACE TWSA has shown great advantages and superiority in flood monitoring

and water resources management for the YRB than traditional flood monitoring approaches.

Furthermore, all these traditional flood monitoring approaches mainly focus on the meteorological conditions or the status of surface water reflecting by various hydro-climatic factors and pay little attention to the importance of antecedent terrestrial water storage conditions before flood events, which can play a critical role in capturing the flood formation processes (Xiong et al., 2021). For example, Reager et al. (2009) applied the TWSA from GRACE data and monthly precipitation to assess the

likelihood for flooding at the regional scale and emphasized the importance of terrestrial water storage signal in the accurate prediction of floods and general runoff. Long et al. (2014) employed the index of flood potential amount using GRACE data and monthly precipitation to investigate hydrological floods and droughts for a large karst plateau in Southwest China and found that higher TWSA estimates are more prone to result in large potential for flooding during rainy season because of the excessive water that cannot be stored further. Therefore, the new proposed index incorporating TWSA can more holistically

quantify the potential of the development of severe floods for regions than common flood potential indices using hydro-climatic observations.

While previous studies have proposed several standardized indices for large-scale flood monitoring based on GRACE-derived TWSA (Chen et al., 2010; Tangdamrongsub et al., 2016), flood monitoring and assessment at sub-monthly time scales remains a challenge using GRACE data due to its coarse temporal resolution (month). Flood monitoring at finer time





scales is pivotal in understanding the regional water cycle under climate change, which ultimately helps to manage the basin-scale water resources effectively and improve the efficiency of early flood warning systems. The application of daily series of TWSA temporally downscaled from GRACE/GRACE-FO satellites data can provide a useful method to comprehensively assess the integrated flood condition considering the changes of both surface and subsurface water storage at sub-monthly time scales. The highest deficit in the temporally downscaled TWSA and the daily precipitation during the wet season, as

revealed by the NDFPI, can indicate the early signs of the region's transition from normal state to a flood-prone situation. Overall, the new proposed NDFPI is proven to be a useful tool for flood monitoring with the finer time scale over large-scale basins, which also makes it possible to monitor extreme flood events timely especially for some regions with limited in-situ streamflow observations.

### 6.3 Uncertainties and limitations

The signals detected by GRACE/GRACE-FO satellites data reflect the changes in regional TWSA under the joint effects of climatic variability and human activates (Xie et al., 2019b). By using the method of linear detrending, long-term trends in series of TWSA estimates have been removed during the reconstruction of TWSA, because they are generally driven by surface conditions and human activities. Human activities such as reservoirs operation, irrigation and water withdrawals cannot be well reconstructed by hydro-climatic factors. Although the detrending method can reduce the impacts of human

activities on reconstructing TWSA to some degree, it could still result in some discrepancies between the results of detrended TWSA and natural TWSA under climatic variability. In future, more attentions should be paid to effectively reconstruct the series of regional TWSA under climatic variability when more detailed information on the statistics of water consumption data induced by human activities are available.

Furthermore, this study presents an effective way to temporally downscale the TWSA estimates from monthly time series

into daily values. This temporal downscaling method is assessed through four case studies across the entire YRB, which could well present the temporal evolution of TWSA at sub-monthly time scales during the wet season. As this study mainly focus on characterizing regional flood potential based on the new proposed NDFPI incorporating temporally downscaled TWSA estimates, we applied this temporal downscaling method on the basin scale. In theory, this method is also suitable for the temporal downscaling of GRACE/GRACE-FO satellites data at the grid cell scale. However, as pointed out by previous

studies (Landerer et al., 2012; Save et al., 2016; Scanlon et al., 2016), gridded TWSA estimates derived from GRACE/GRACE-FO satellites data involve relatively large uncertainty induced by associated measurement errors and signal leakage errors. As a result, the accuracy of TWSA estimates can ultimately exert a direct influence on the optimized parameter sets that are obtained for trained models in each grid cell, which is a contributing factor of the uncertainty. In addition, the forcing data of these models used for temporal downscaling, including air temperature, precipitation and

GLDAS Noah derived SMSA, may also contain some errors and uncertainties due to the uneven spatial distribution of



meteorological stations and natural measurement errors (Lv et al., 2017). These errors and uncertainties from the input data could be propagated into the machine learning-based models (e.g. MLP model), resulting in a broad range of differences between the observations and the simulated results. The latest study has noticed the importance of spatially correlated features and made some initial attempts to make full use of the spatially correlated features associated with images for

predictions based on the convolutional neural network (CNN) model with the goal of providing more accurate TWSA estimates (Mo et al., 2022). Therefore, a thorough consideration of the spatiotemporally correlated features among each grid cell will be taken in our future work when downscaling the TWSA estimates from monthly time series to daily time series at the grid cell scale and fully understand the complex underlying mechanism for TWSA variations during the wet season.

## 7 Conclusions

In the present study, we downscaled the GRACE/GRACE-FO derived TWSA estimates from monthly time series to daily time series in the YRB by establishing a relationship between TWSA estimates and hydro-climatic factors based on machine learning techniques. Furthermore, the temporally downscaled TWSA data combined with daily precipitation were adopted to monitor the extreme flood events over the entire YRB in 2020. The main conclusions can be drawn as follows:

(1) When reconstructing monthly TWSA in the YRB, the MLP model shows the best performance with RMSE = 10.9

mm/month, NSE = 0.89, the MLR model follows with RMSE =13.4 mm/month, NSE = 0.84, and the LSTM model shows the lowest performance with RMSE = 15.1 mm/month, NSE = 0.81 during the validation period;

(2) Based on the MLP model, monthly time series of TWSA were temporally downscaled to daily data by using meteorological observations and outputs from a land surface model. The results showed highly consistency with original monthly TWSA estimates derived from GRACE/GRACE-FO satellites data with regard to seasonal cycles;

(3) By jointly using daily average precipitation anomalies and temporally downscaled TWSA, the proposed NDFPI can effectively detect the flood events at sub-monthly time scales occurred in summer 2020 for the entire YRB;

(4) The comparison analysis indicates that the flood events can be monitored by the proposed NDFPI earlier than traditional streamflow observations with respect to the YRB and its individual basins, which is very vital for flood forecasts and warning across this region.

Overall, the present study shows the great potential of temporally downscaled GRACE/GRACE-FO satellites data in a wide range of hydrological applications, such as monitoring the extreme flood events. The study provides an effective means for the temporal downscaling of original TWSA estimates from GRACE/GRACE-FO satellites data and will help facilitate the sustainable management of water resources and develop monitoring and early warning systems for severe flood events over large-scale basins.



**Author Contributions**


Yue-Ping Xu and Jingkai Xie designed the study; Yue-Ping Xu guided the research and revised the manuscript; Jingkai Xie did the main calculations and wrote the draft of the manuscript; Hongjie Yu and Yan Huang performed data pre-processing; Yuxue Guo helped to process the raw GRACE data.

**Acknowledgements**

The authors would like to thank both China Meteorological Administration and Yangtze River Conservancy Commission of Ministry of Water Resources (http://www.cjw.gov.cn/) for providing the meteorological observations and hydrological data used in this study. We sincerely thank the NASA MEaSUREs Program and the Center for Space Research for providing GRACE/GRACE Follow-On JPL and CSR Level 3 Release 6 data, both of which are available from https://podaac.jpl.nasa.gov. We also sincerely thank the Goddard Earth Sciences (GES) Data and Information Services
Center (DISC) for providing soil moisture storage acquired from the Global Land Data Assimilation System (GLDAS) data (https://disc.sci.gsfc.nasa.gov/).

**Financial support**

This study is financially sponsored by the National Natural Science Foundation of China (52109037, 52009121), Zhejiang Key Research and Development Program (2021C03017), and the Fundamental Research Funds for the Zhejiang Provincial
Universities (2021XZZX015).

**Competing interests**

The authors declare that they have no conflict of interest.

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



**Table 1: A summary of relevant literature on monitoring extreme flood events using GRACE/GRACE-FO data. GRACE = Gravity Recovery and Climate Experiment mission; GRACE-FO = Gravity Recovery and Climate Experiment Follow-On mission; GLDAS = Global Land Data Assimilation system; TRMM = Tropical Rainfall Measuring Mission; MODIS = Moderate-Resolution Imaging Spectroradiometer.**

| Study | Study region | Source data | Period | Temporal resolution | Main contributions |
|---|---|---|---|---|---|
| Chen et al. (2010) | Amazon basin | GRACE RL04 data; precipitation | 2002 to 2009 | Month | Measuring large-scale extreme flood events |
| Long et al. (2014) | Yun-Gui Plateau | GRACE RL05 data; hydrometeorological data | 2003 to 2012 | Month | Evaluating the frequency and severity of droughts and floods over the regions |
| Reager et al. (2014) | Mississippi River basin | GRACE data; GLDAS data; stream gauge data | 2003 to 2011 | Month | Characterizing regional flood potential and assessing the predisposition of a river basin to flooding |
| Tangdamrongsub et al. (2016) | Tonlé Sap basin | GRACE RL05 data; TRMM; MODIS; hydrological model | 2002 to 2014 | Month | Quantifying the flood events at both basin and sub-basin scales |
| Chen et al. (2018) | Liao River basin | GRACE RL05 data; meteorological data; hydrological model | 2002 to 2016 | Month | Monitoring the drought and flood patterns based on the total storage deficit index |
| Yang et al. (2021) | Yangtze River Basin | GRACE/GRACE-FO RL06 data; meteorological data; teleconnection indices | 2002 to 2018 | Month | Investigating the flood risk factors and analyzing the impact of climate change factors on flood events |
| Shah et al. (2021) | Indian subcontinent | GRACE RL06 data; meteorological data | 2002 to 2016 | Month | Examining the role of changes in terrestrial water and groundwater storage on flood potential |
| This study | Yangtze River Basin | GRACE/GRACE-FO RL06 data; runoff; meteorological data | 2003 to 2020 | Day | Monitoring the evolution of extreme flood events based on temporally downscaled GRACE data |



**Table 2: An overview of all datasets used in this study.**

| Data | Source | Temporal resolution | Spatial resolution | Time span |
|---|---|---|---|---|
| Terrestrial water storage | GRACE/GRACE-FO CSR | Month | 0.5° | 2002 - Now |
| anomaly (TWSA) | GRACE/GRACE-FO JPL | Month | 0.5° | 2002 - Now |
| | GRACE/GRACE-FO GSFC | Month | 0.5° | 2002 - Now |
| Soil moisture storage (SMS) | GLDAS 2.1 - Noah | 3 hours | 1° | 2002 - Now |
| Precipitation (T) | CMA | Day | / | 2003 - 2020 |
| Temperature (P) | CMA | Day | / | 2003 - 2020 |
| Streamflow | In situ | Day | / | 2003 - 2020 |

Note. GRACE = Gravity Recovery and Climate Experiment mission; GRACE-FO = Gravity Recovery and Climate Experiment Follow-On mission; CSR = Center for Space Research; JPL = Jet Propulsion Laboratory; GSFC = Goddard Space Flight Center; GLDAS = Global Land Data Assimilation system; CMA = China Meteorological Administration.



**Table 3: Performances of different models in simulating monthly TWSA across the YRB during 2003-2020.**

| Scenarios | | MLP (RMSE/NSE) | LSTM (RMSE/NSE) | MLR (RMSE/NSE) |
|---|---|---|---|---|
| Scenario 1 | 2003/01 - 2014/06 (Training) (70%) | 10.71/0.89 | 12.14/0.86 | 11.63/0.87 |
| | 2014/08 - 2020/12 (Validation) (30%) | 26.12/0.50 | 26.62/0.15 | 24.32/0.57 |
| | 2003/01 - 2020/12 (Overall) (100%) | 16.91/0.76 | 17.74/0.68 | 16.51/0.77 |
| Scenario 2 | 2005/06 - 2018/06 (Training) (70%) | 13.54/0.84 | 14.61/0.79 | 14.33/0.82 |
| | 2003/01 - 2005/05 and 2018/07 - 2020/12 (Validation) (30%) | 23.32/0.59 | 25.42/0.17 | 20.14/0.70 |
| | 2003/01 - 2020/12 (Overall) (100%) | 17.15/0.75 | 18.57/0.65 | 16.21/0.78 |
| Scenario 3 | 2007/10 - 2020/12 (Training) (70%) | 15.76/0.80 | 17.84/0.68 | 17.24/0.76 |
| | 2003/01 - 2007/09 (Validation) (30%) | 10.92/0.89 | 15.12/0.81 | 13.41/0.84 |
| | 2003/01 - 2020/12 (Overall) (100%) | 14.41/0.83 | 17.61/0.73 | 16.11/0.78 |

Note. TWSA = Terrestrial water storage anomalies; YRB = Yangtze River Basin; MLP = Multi-layer perceptron neural network; LSTM = Long short-term memory network; MLR = Multiple linear regression. 70%, 30% and 100% represent the corresponding proportions to all samples in the training, the validation and the entire periods respectively. RMSE/NSE represent the Root mean square error (mm/month) and Nash-Sutcliffe efficiency coefficient between the simulated TWSA with the observed TWSA respectively. Noted that GRACE/GRACE-FO derived TWSA in some months are not available
due to the problem of battery management.

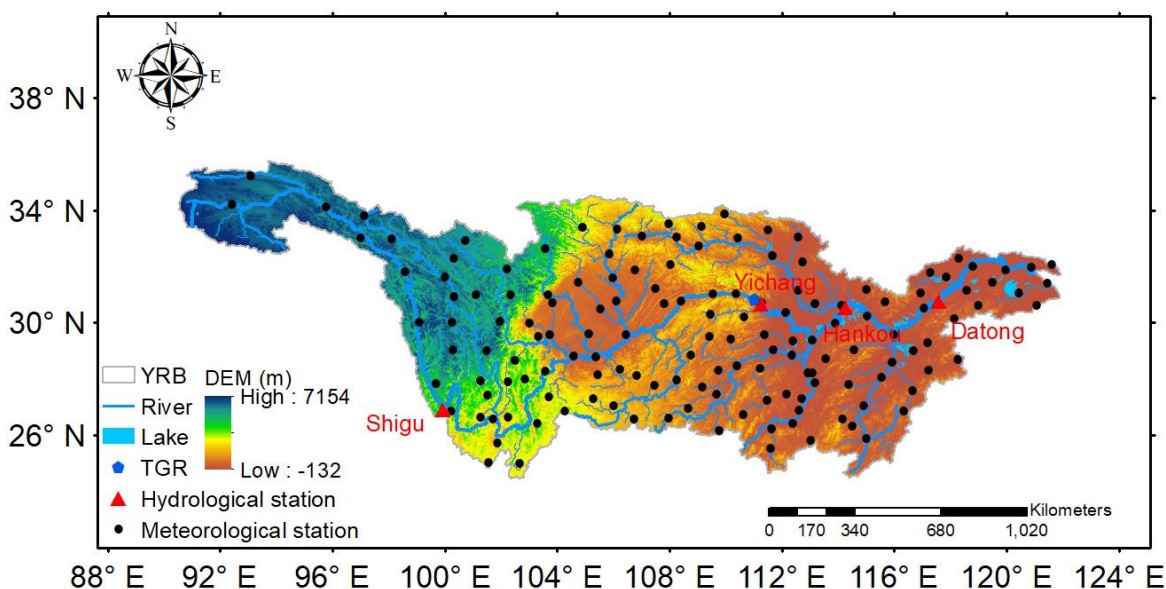

**Figure 1: Location of the Yangtze River Basin (YRB) in China and its topography. Distribution of meteorological stations and hydrological stations are also shown in this figure. TGR = Three Gorges Reservoir; DEM = Digital Elevation Model.**

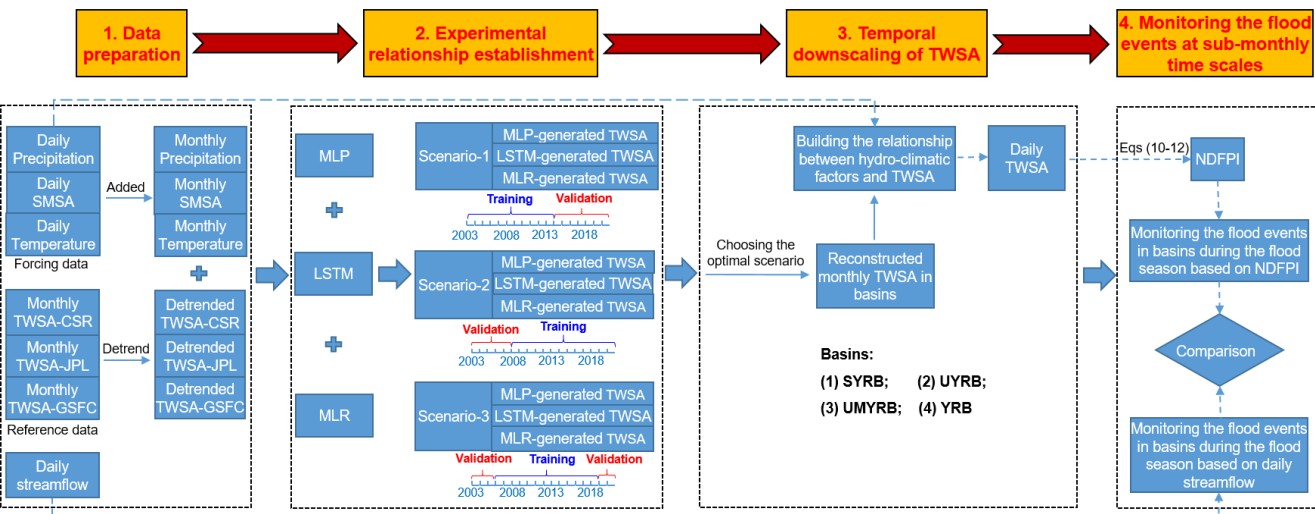

**Figure 2: A detailed flow diagram illustrating the temporal downscaling of GRACE/GRACE-FO derived TWSA. GRACE = Gravity Recovery and Climate Experiment mission; GRACE-FO = Gravity Recovery and Climate Experiment Follow-On mission; SMSA = Soil moisture storage anomaly; TWSA = Terrestrial water storage anomaly; CSR = Center for Space Research; JPL = Jet Propulsion Laboratory; GSFC = Goddard Space Flight Center; SYRB = Source regions of the Yangtze River Basin; UYRB = Upper regions of the Yangtze River Basin; UMYRB = Upper and the Middle regions of the Yangtze River Basin; YRB = Yangtze**
**River Basin; MLP = Multi-layer perceptron neural network; LSTM= Long short-term memory; MLR = Multiple linear regression; NDFPI = Normalized daily flood potential index.**





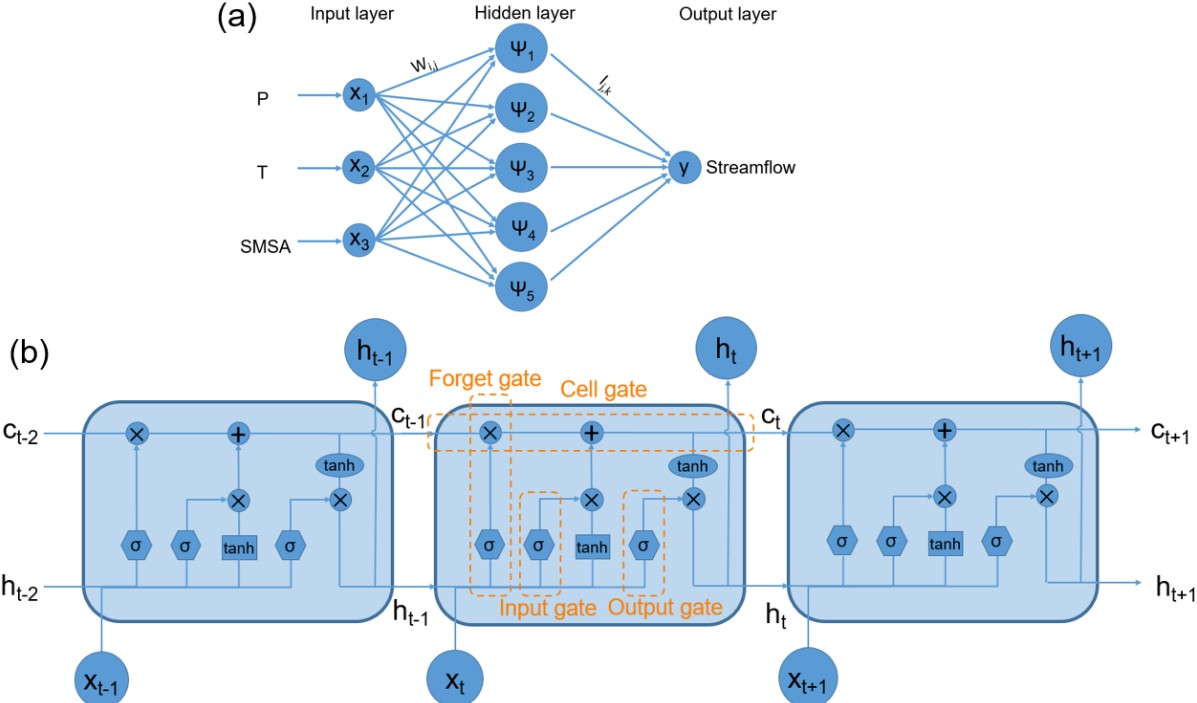

**Figure 3: Architecture of (a) a typical three-layer multi-layer perceptron (MLP) neural network and (b) a typical a long-short time memory (LSTM) network. $\Psi_i$ denotes the sigmoid transfer function, $W_{i,j}$ represent connection weights between the input layer and the hidden layer, $I_{j,k}$ represent connection weights between the hidden layer and the output layer. $x_t$, $c_t$, and $h_t$ represent the standardized input variable, hidden gate and cell gate at the current time t.**

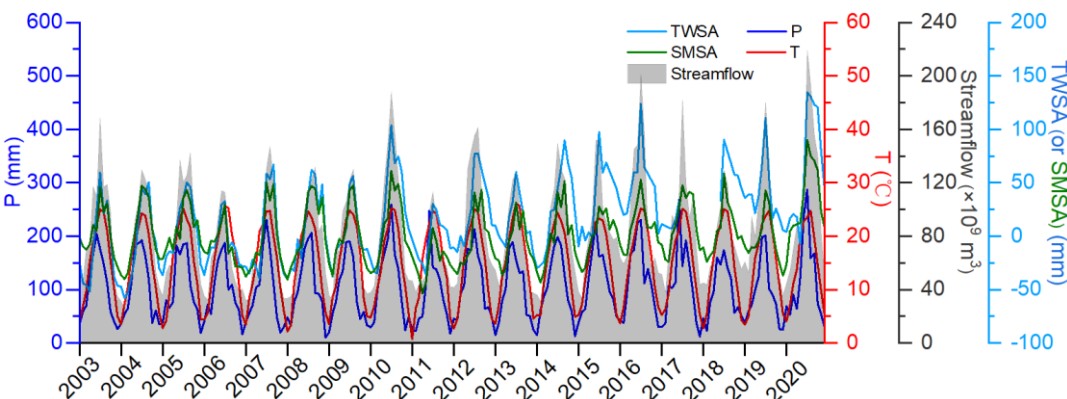

**Figure 4: Monthly time series of precipitation (P, mm), temperature (T, °C), terrestrial water storage anomaly (TWSA, mm), soil moisture storage anomaly (SMSA, mm) and streamflow (×10⁹ m³) across the YRB during 2003-2020. Streamflow data is obtained at the Datong hydrological station (shown in Figure 1). YRB = Yangtze River Basin.**





**Figure 5: Comparison between monthly TWSA derived from GRCACE/GRACE-FO satellites data (observation) and that simulated by different models (validation) for (a-c) the SYRB, (d-f) the UYRB, (g-i) the UMYRB and (j-l) the YRB respectively during 2003-2020 with showing statistics of the comparison including root mean square errors (RMSE) (mm/month) and Nash-Sutcliffe efficiency (NSE). Note that TWSA shown in this figure are detrended because hydro-climatic factors may not fully simulate all the long-term trends. The models showing the best performance in simulating TWSA during the validation periods have been bold for each region. SYRB = Source regions of Yangtze River Basin; UYRB = Upper regions of Yangtze River Basin; UMYRB = Upper and middle regions of Yangtze River Basin; YRB = Yangtze River Basin.**



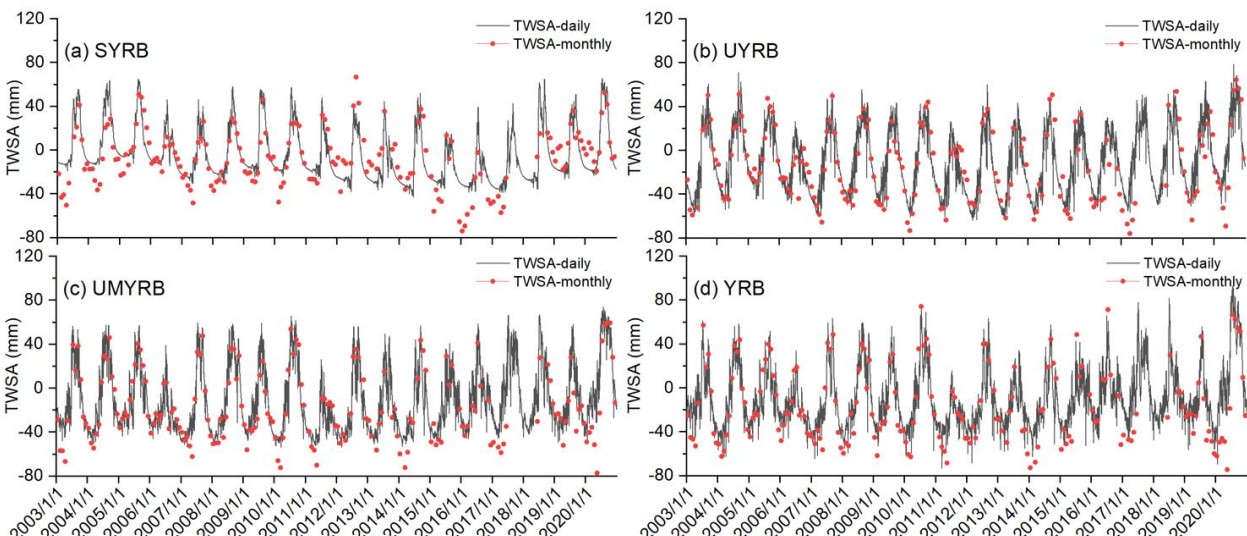

**Figure 6: Daily time series of TWSA temporally downscaled by the MLP model (TWSA-daily, represented by grey lines) for (a) the SYRB, (b) the UYRB, (c) the UMYRB and (d) the YRB respectively during 2003-2020. Note that monthly TWSA estimates derived from GRACE/GRACE-FO satellites data (TWSA-monthly, represented by red dots) shown in this figure are detrended because hydro-climatic factors may not fully simulate their long-term trends. TWSA = Terrestrial water storage anomaly; MLP = Multi-layer perceptron neural network; SYRB = Source regions of Yangtze River Basin; UYRB = Upper regions of Yangtze River Basin; UMYRB = Upper and middle regions of Yangtze River Basin; YRB = Yangtze River Basin.**



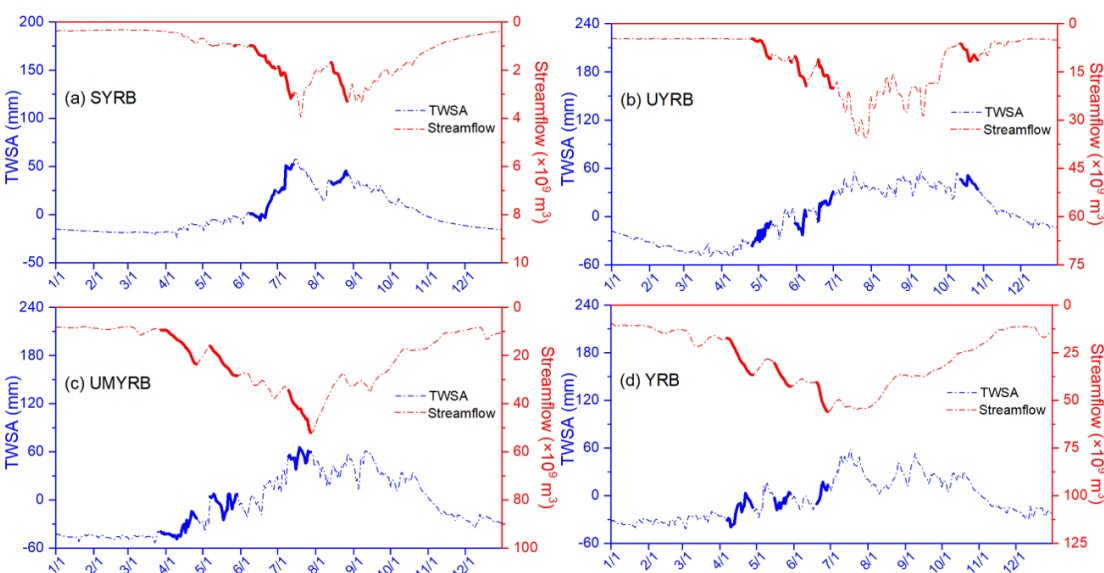

**Figure 7: Daily TWSA temporally downscaled by the MLP model versus streamflow during flood events across (a) the SYRB, (b) the UYRB, (c) the UMYRB and (d) the YRB respectively in 2010. The bold blue dash lines and bold red dash lines represent daily TWSA and streamflow during the period between the beginning and end of each runoff event. TWSA = Terrestrial water storage anomaly; MLP = Multi-layer perceptron neural network; SYRB = Source regions of Yangtze River Basin; UYRB = Upper regions of Yangtze River Basin; UMYRB = Upper and middle regions of Yangtze River Basin; YRB = Yangtze River Basin.**



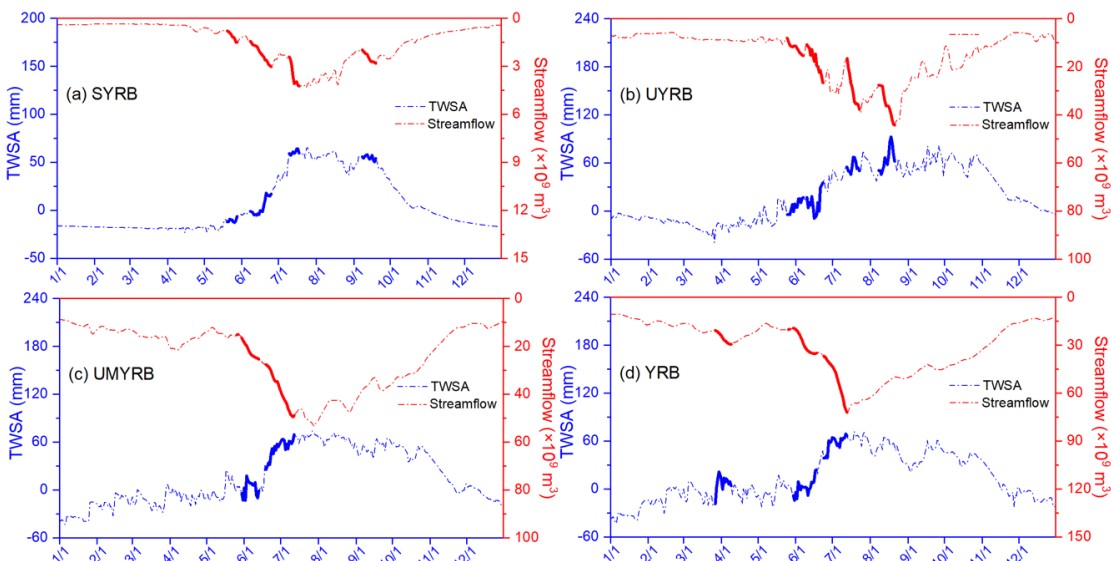

**Figure 8: Same as Figure 7 but in 2020.**



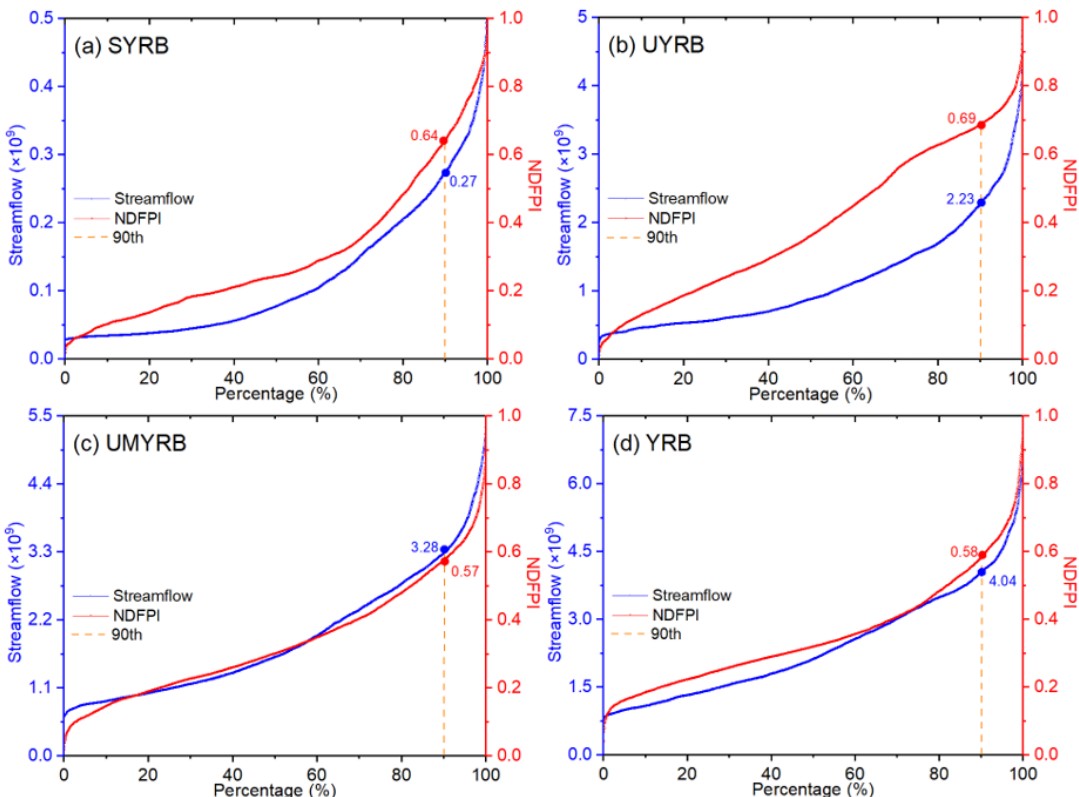

**Figure 9: Percentile duration curves of daily streamflow observations and NDFPI for the 90th percentile floods across (a) the SYRB, (b) the UYRB, (c) the UMYRB and (d) the YRB respectively during 2003-2020. The red dots and blue dots represent threshold values of daily streamflow and NDFPI for the 90th percentile floods across different regions. SYRB = Source regions of Yangtze River Basin; UYRB = Upper regions of Yangtze River Basin; UMYRB = Upper and middle regions of Yangtze River Basin; YRB = Yangtze River Basin.**

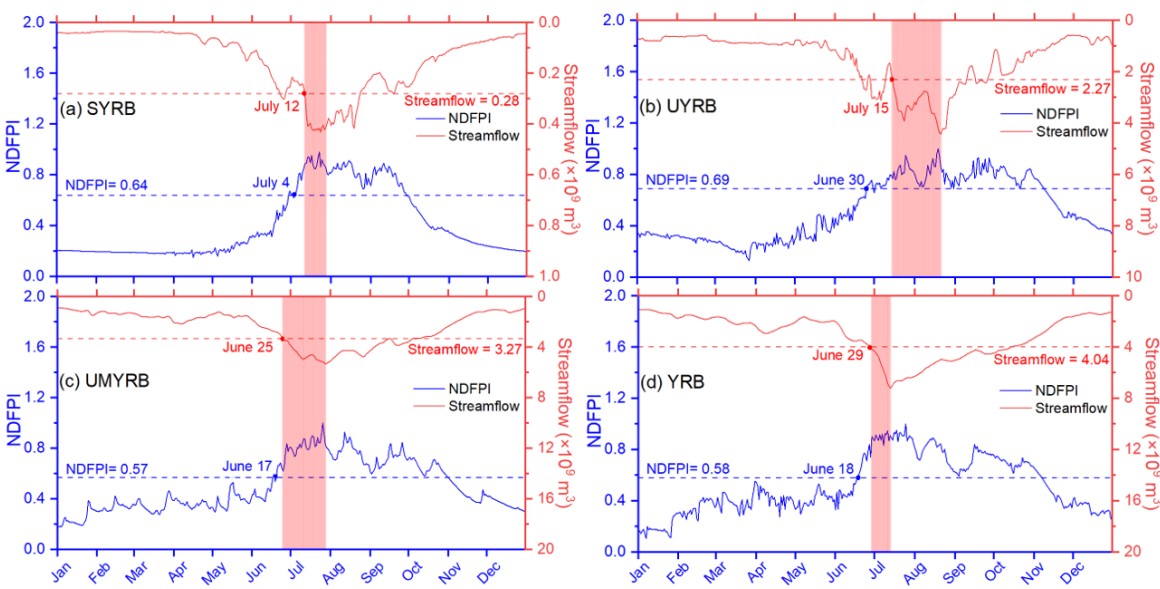

815

**Figure 10: Comparison between basin averaged NDFPI and daily streamflow observations for the 90th percentile floods in summer 2020 across (a) the SYRM (observed at Shigu station), (b) the UYRB (observed at Yichang station), (c) the UMYRB (observed at Hankou station) and (d) the YRB (observed at Datong station). Pink rectangles denote the duration period between the thresholds of daily streamflow for the 90th percentile floods and peak streamflow observed at the controlling hydrological stations over different regions. The thresholds of daily streamflow and NDFPI for the 90th percentile floods are represented by the red dash lines and blue dash lines respectively. Note that the scales of streamflow shown in each figure are not always same. SYRB = Source regions of Yangtze River Basin; UYRB = Upper regions of Yangtze River Basin; UMYRB = Upper and middle regions of Yangtze River Basin; YRB = Yangtze River Basin.**