# Peer review of "Monitoring the extreme flood events in the Yangtze River Basin based on GRACE/GRACE-FO satellite data"

_Hydrology and Earth System Sciences, 2022_

## Author Comment (AC1)

**To anonymous Referee #1, 23 May 2022**

General comments for the authors' reference:

This manuscript is about the temporal downscaling of satellite based terrestrial water storage anomalies (TWSA) using data-based methods with in-situ meteorological data and assimilated soil moisture data as inputs. In addition, a new index is proposed incorporating downscaled TWSA estimates and daily precipitation to monitor flood events. The concepts are applied to four (sub-)basins of the Yangtze River Basin (YRB) in China.

Overall, the paper is well structured and written, and presents an interesting approach to temporally downscale GRACE/ GRACE-FO data to obtain daily TWSA estimates and its subsequent use in a daily normalized flood index. The authors generally use informative figures to illustrate their results. However, several issues need attention such as the concept and method to remove trends from the TWSA time series (see specific comment 7), the assumption that the data-based relationships between TWSA and hydro-meteorological variables are time scale invariant (see specific comment 16) and the advantages of the proposed method and possible operationalisation (see specific comment 21). These and other specific and technical comments can be found below.

Response: Thanks for your valuable suggestions. We have tried our best to improve the manuscript both in the language and contents based on your comments and suggestions. We sincerely appreciate your valuable inputs and we hope that the changes implemented are sufficient for acceptance of our paper for publication.

Specific comments
(1) L23: Only damage to structures and agriculture? No other types of flood damage and/ or casualties? See also line 38, where only a monetary value is mentioned without non-monetary damage and casualties/victims.
Response: As described in Line 23 and Line 64, we have rewritten these sentences to provide more information about the effects of extreme floods based on your suggestions.

"Extreme floods, as one of the most destructive natural hazards, not only cause lots of casualties in China and around the world, but also have considerably wider and adverse economic consequences (Dottori et al., 2018)."

"For example, in Year 2020, the YRB has experienced one of the most extreme flood events on record. According to the data from the Ministry of Emergency Management of the People's Republic of China, a total of 38.173 million people were affected and 27,000 houses collapsed due to the 2020 flood, with 56 deaths or disappearances and a great economic loss of $27.68 billion (USD) (Jia et al., 2021)."

**References**

Dottori, F., Szewczyk, W., Ciscar, J., Zhao, F., Alfieri, L., Hirabayashi, Y., Bianchi, A., Mongelli, I., Frieler, K., Betts, R.A., Feyen, L., 2018. Increased human and economic losses from river flooding with anthropogenic warming. Nat. Clim. Change. 8, 781-786.

Jia, H., Chen, F., Pan, D., Du, E., Wang, L., Wang, N., Yang, A., 2021. Flood risk management in the Yangtze River basin - Comparison of 1998 and 2020 events. Int. J. Disaster Risk Reduct. 68, 102724.

(2) L31: Why does this study use the YRB as case study? How representative is the YRB for other basins in the world? To what extent can the methods and/ or results be generalized to other basins? This needs more attention in the manuscript, both in the introduction and discussion sections. The introduction of the YRB might be moved to the end of the introduction section (after objective) to emphasize the role of the YRB being a case study to show the concepts and methods of this paper.

Response: Thanks for your valuable suggestions. More attentions have been paid to explain the reasons why we use the YRB as case study in the manuscript, both in the introduction and discussion sections based on your suggestions. As described in the **Section of Introduction**, the reasons why we use the YRB as case study can be explained by the flowing two aspects. On one hand, **as described in Line 57 to Line 58**, the YRB is one of the most important basins in China, because it can provide hundreds of millions of people with drinking water, food, hydropower, and other ecosystem services. On the other hand, **as described in Line 58 to Line 67**, the YRB has been regarded as one of the most sensitive and vulnerable regions that suffered from severe extreme floods. During the past decades, extreme floods have caused catastrophic damage to human beings in this basin, such as the 1998 Yangtze River Flood. Meanwhile, it has been found that both the frequency and severity of extreme flood events generally showed an upward trend in the YRB in the recent decades (Huang et al., 2015; Liu et al., 2019; Yang et al., 2021; Zhang et al., 2008). Therefore, there is a great need to provide more deep insights into the monitoring of extreme floods in the YRB.

Furthermore, we also added some sentences to explain the reasons why we use the YRB as case study in the **Section of Discussion**. **As described in Line 544 to Line 556**, the present study shows the great potential of temporally downscaled GRACE/GRACE-FO satellite data in monitoring the extreme flood events. Meanwhile, the study provides an effective means for the temporal downscaling of original TWSA estimates from GRACE/GRACE-FO satellite data, which can help develop monitoring and early warning systems for severe flood events over large-scale basins. Therefore, the methods and/ or results shown in this study can also provide important implications of flood hazard prevention and water resource management for other similar basins that are prone to suffer from severe extreme floods.

65 Furthermore, the introduction of the YRB has been moved to **the end of the introduction section** (after objective) to emphasize the role of the YRB being a case study to show the concepts and methods of this paper as you suggested (Line 57 to Line 67).

"The YRB is one of the most important basins in China, because it can provide freshwater, hydropower, food, and other ecosystem services for hundreds of millions of people. Meanwhile, the YRB has been regarded as one of the most sensitive and vulnerable regions that suffered from severe floods due to its highly uneven rainfall pattern (Zhang et al., 2021). During the past decades, the increasingly intensified human activities and climate change have substantially changed the hydrological cycle in the YRB and thus accelerated the variation of flood characteristics in this region (Fang et al., 2012; Wang et al., 2011). It has been found that both the frequency and severity of extreme flood events generally showed an upward trend in the YRB in recent decades owing to substantial changes in climate, infrastructure and land use (Huang et al., 2015; Liu et al., 2019; Yang et al., 2021; Zhang et al., 2008). For example, in Year 2020, the YRB has experienced one of the most extreme flood events on record. According to the data from the Ministry of Emergency Management of the People's Republic of China, a total of 38.173 million people were affected and 27,000 houses collapsed due to the 2020 flood, with 56 deaths or disappearances and a great economic loss of \$27.68 billion (USD) (Jia et al., 2021)."

(3) L86-88: Which ranges are meant here? Ranges in space or in time (i.e. interannual variability)? Both temperature and precipitation ranges seem a bit narrow for a large river basin like YRB.

Response: As described in Line 89 to Line 91, the results shown in this study refer to ranges in time, which are derived from the interannual variability of the precipitation and temperature during the study period. As stated in this study, precipitation and temperature from 2003-2020 are provided by the China Meteorological Administration (CMA) (http://data.cma.cn/) with a total of 150 National Meteorological Observatory stations distributed in the YRB (shown in Fig. 1). The reason why both temperature and precipitation ranges seem a bit narrow for a large river basin like YRB might be that the study period spanning from 2003 to 2020 is relatively limited in this study and the ranges are basin-averaged values. For example, according to the statistics from the Water Resources Bulletin of the Yangtze River (http://cjw.gov.cn/zwzc/bmgb/), mean annual precipitation in the YRB shows a narrow range from 931 mm to 1282 mm from 2006 to 2020, which is highly consistent with the results shown in this study.

"According to the observations from meteorological stations, the mean annual air temperature of this basin ranges from 14.4 ℃ to 15.4 ℃ and mean annual precipitation ranges from 1049 mm to 1424 mm during the study period 2003-2020."

(4) L104-106: Why did the authors use the average of three types of GRACE and GRACE-FO solutions? Do these three types show equal performance in estimating TWSA? If not, wouldn't it be better to use weights depending on individual product performance when averaging the three products?

Response: Many studies (Ali et al., 2021; Liu et al., 2020; Liu et al., 2021) indicate that these GRACE
100  and GRACE-FO solutions show equal performance in estimating TWSA and also directly use the average
of these GRACE and GRACE-FO solutions to estimate the variation of regional TWSA, which is
consistent with the method applied in this study. For example, the study of Seyoum (2018) clearly
indicates that basin-wide TWSA could be calculated by averaging multiple GRACE and GRACE-FO
solutions over the basin in order to reduce uncertainty. Furthermore, no other in-situ observations are
105  available for deriving the weights depending on individual product performance. Therefore, we use the
average of three types of GRACE and GRACE-FO solutions to characterize the variations of TWSA in
the YRB in this study.

**References**

Ali, S., Wang, Q., Liu, D., Fu, Q., Rahaman, M.M., Faiz, M.A., Cheema, M.J.M., 2021. Estimation of
110    spatio-temporal groundwater storage variations in the lower transboundary Indus basin using
GRACE satellite. J. Hydrol. 127315.

Liu, X., Feng, X., Ciais, P., Fu, B., Hu, B., Sun, Z., 2020. GRACE satellite-based drought index indicating
increased impact of drought over major basins in China during 2002-2017. Agric. For. Meteorol.
291, 108057.

115  Liu, B., Zou, X., Yi, S., Sneeuw, N., Cai, J., Li, J., 2021. Identifying and separating climate- and human-
driven water storage anomalies using GRACE satellite data. Remote Sens. Environ. 263, 112559.

Seyoum, W.M., 2018. Characterizing water storage trends and regional climate influence using GRACE
observation and satellite altimetry data in the Upper Blue Nile River Basin. J. Hydrol. 566, 274-284.

(5) L119-120: Is the Thiessen method the most suitable method for spatial averaging given for instance
120  elevation differences in (sub-)basins?

Response: In this study, the classic Thiessen polygon method is applied to calculate the basin-average
precipitation based on the observations from different meteorological stations. In fact, many studies (Chao
et al., 2021; Spencer et al, 2019; Strauch et al., 2012) have calculated the catchment-averaged
precipitation in different basins including the Yangtze River Basin. For example, Chen et al., (2014)
125  calculated the basin-average precipitation for the Yangtze River Basin and its sub-catchments using the
Thiessen polygon method to spatially distribute the single point records of raingauges. Wang et al., (2019)
calculated basin-wide precipitation for the upper Yangtze River basin and its sub-catchments where
obvious elevation differences can be found using the Thiessen polygon method. Li et al., (2018) also
calculated basin-wide precipitation and temperature the upper Yangtze River basin and its sub-catchments
130  using the Thiessen polygon method. This method is a classical method used for calculating basin

precipitation. In our future study, other methods can be investigated to consider the elevation differences, in particular in the upper part of the YRB.

**References**

Chao, L., Zhang, K., Yang, Z. L., Wang, J., Lin, P., Liang, J., Li, Z., Gu, Z., 2021. Improving flood simulation capability of the WRF-Hydro-RAPID model using a multi-source precipitation merging method. J. Hydrol. 592, 125814.

Chen, J., Wu, X., Finlayson, B.L., Webber, M., Wei, T., Li, M., Chen, Z., 2014. Variability and trend in the hydrology of the Yangtze River, China: Annual precipitation and runoff. J. Hydrol. 513, 403-412.

Li, D., Lu, X.X., Yang, X., Chen, L., Lin, L., 2018. Sediment load responses to climate variation and cascade reservoirs in the Yangtze River: a case study of the Jinsha River. Geomorphology 322, 41-52.

Spencer, S.A., Silins, U., Anderson, A.E., 2019. Precipitation-Runoff and storage dynamics in watersheds underlain by till and permeable bedrock in Alberta's Rocky Mountains. Water Resour. Res. 55(12), 10690-10706.

Strauch, M., Bernhofer, C., Koide, S., Volk, M., Lorz, C., Makeschin, F., 2012. Using precipitation data ensemble for uncertainty analysis in SWAT streamflow simulation. J. Hydrol. 414, 413-424.

Wang, W., Zhu, Y., Dong, S., Becker, S., Chne, Y., 2019. Attribution of decreasing annual and autumn inflows to the Three Gorges Reservoir, Yangtze River: Climate variability, water consumption or upstream reservoir operation? J. Hydrol. 579, 124180.

(6) L143: Shouldn't the 3-hourly SMS data be summed instead of multiplied with 8? The SMS values for each 3-hourly time step are not necessarily the same, so multiplication with 8 will result in errors.

Response: Thanks for your valuable suggestions. As described Line 141 to Line 150, we have carefully checked and rewritten this section to better present the process of collecting SMS data. According to Table 2, the temporal resolution of SMS derived from GLDAS Noah land surface model is three hours. That is, there are eight SMS outputs for a specific day using the GLDAS Noah land surface model. Therefore, daily SMS estimates can be obtained by estimating the average of eight SMS outputs included in one day, because soil moisture storage is a state variable.

"Therefore, in this study we adopt the SMS (kg/m$^2$) with a spatial resolution of $0.25\,° \times 0.25\,°$ from the Global Land Data Assimilation System version 2.1 (GLDAS 2.1) Noah land surface model to estimate their correlations with regional TWSA derived from the GRACE and GRACE-FO satellite data. This product can provide the simulations of SMS at four different depths of soil layers from 0 to 200 cm, that is, 0 - 10 cm, 10 - 40 cm, 40 - 100 cm and 100 - 200 cm depths per three hours. To keep consistent with TWSA, the original value of SMS should be transferred into soil moisture storage anomaly values (SMSA) after subtracting the time-mean baseline during the period of 2004-2009. Furthermore, the temporal resolution of original SMS derived from GLDAS 2.1 Noah land surface model can be decreased from 3-hours to 1-day and 1-month composite respectively, which is consistent with the methods applied in previous studies (Mohanasundaram et al., 2021; Mulder et al., 2015; Syed et al., 2008).

**References**

Mohanasundaram, S., Mekonen, M.M., Haacker, E., Ray, C., Lim, S., Shrestha, S., 2021. An application of GRACE mission datasets for streamflow and baseflow estimation in the Conterminous United States basins. J. Hydrol. 601, 126622.

Mulder, G., Olsthoorn, T.N., Al-Manmi, D.A.M.A., Schrama, E.J.O., Smidt, E.H., 2015. Identifying water mass depletion in northern Iraq observed by GRACE. Hydrol. Earth Syst. Sci. 19 (3), 1487-1500.

Syed, T.H., Famiglietti, J.S., Rodell, M., Chen, J., Wilson, C.R., 2008. Analysis of Terrestrial Water Storage Changes from GRACE and GLDAS. Water Resour. Res. 44, W02433.

(7) L175-177: It is not completely clear which trends are considered here. As the authors indicate, long-term trends due to human activities should be removed in order to analyse the relation between 'natural' TWSA and hydroclimatic variables. The question is whether one can remove the human induced trend without removing part of a possible hydroclimatic (natural) induced trend. Did the authors analyse trends in precipitation and evapotranspiration (or temperature) to see whether climatic trends also could have been partly responsible for trends in TWSA? Should SMSA also be detrended similarly to TWSA? And did all relevant human activities cause a trend in TWSA or also abrupt changes in TWSA? Finally, it is doubtful whether the trend is always linear and hence linear methods can be used to remove the trends.

Response: On one hand, the effects of human activities on changes in TWSA cannot be neglected considering that the YRB is strongly influenced by various human activities, such as the construction of Three Gorges Reservoir and intense human water consumption (Huang et al., 2015; Yao et al., 2021). On the other hand, detailed statistics related to human use such as water consumption, reservoir operation and inter-basin water diversion projects are not available in this region. Therefore, we have removed the linear trends from the original time series of TWSA in training and calibration periods assuming that all

these linear trends arise from the joint effects of various human activities, such as the water withdrawals and reservoir operation over the study region. This application is consistent with many previous studies when reconstructing TWSA estimates (Bai et al., 2022; Liu et al., 2021; Rodell et al., 2018; Yang et al., 2021).

Furthermore, we also analyzed monthly times series of precipitation and evapotranspiration (shown in Figure R1) to investigate whether climatic trends also could have been partly responsible for trends in TWSA based on your suggestions. To accurately characterize the monthly variations in ET in the YRB, one high-resolution ET product is applied in this study, namely the Global Land Evaporation Amsterdam Model Version 3.5a (GLEAM v3.5a) product with a 0.25 °×0.25 °resolution (Miralles et al. 2011; Martens et al. 2017) (https://www.gleam.eu/). This ET product has been widely used in many studies (Baik et al., 2018; Khan et al., 2018; Liu et al., 2022; Rehana et al., 2021) due to its high accuracy and resolution. As shown in Figure R1, it seems that no significant trends can be found in monthly precipitation during the study period for the YRB. Meanwhile, a slight increase trend can be found in monthly evapotranspiration but this trend is not significant. The variable of SMSA is simulated by Noah land surface model by compiling ground and satellite-based observations while no data reflecting human activities are incorporated or assimilated in this model for the YRB (Rodell et al., 2004). That is, SMSA can be viewed as the outputs from the land surface model under the sole effects of climatic variability. Therefore, we do not have to remove the linear trends of SMSA when reconstructing TWSA estimates. Nevertheless, considering that this method might be further applied in other regions where substantial changes in climate occurred, we also added some sentences in Section 6.3, showing the possible uncertainties and limitations of this method applied in this study as you suggested.

"By using the method of linear detrending, long-term trends in series of TWSA estimates have been removed during the reconstruction of TWSA, because they are generally driven by various human activities such as irrigation and reservoir operation and water withdrawals, all of which cannot be well reconstructed by hydro-climatic factors (Humphrey and Gudmundsson, 2019). Although the detrending method can reduce the impacts of human activities on reconstructing TWSA to some degree, it could still result in some discrepancies between the results of detrended TWSA and natural TWSA under climatic variability particularly in some regions where intense human activities existed. In future, more attentions should be paid to reconstruct the series of regional TWSA under climatic variability when more detailed statistics related to human use such as water consumption, reservoir operation and inter-basin water diversion projects are available."

[Figure]

**Figure R1. Monthly times series of precipitation and evapotranspiration in the YRB during 2003-2020.**

**References**

Bai, H., Ming, Z., Zhong, Y., Zhong, M., Kong, D., Ji, B., 2022. Evaluation of evapotranspiration for exorheic basins in China using an improved estimate of terrestrial water storage change. J. Hydrol. 610, 127885.

Baik, J., Liaqat, U.W., Choi, M., 2018. Assessment of satellite- and reanalysis-based evapotranspiration products with two blending approaches over the complex landscapes and climates of Australia. Agric. Water Manag. 263, 388-398.

Huang, Y., Salama, M.S., Krol, M.S., Su, Z., Hoekstra, A.Y., Zeng, Y., Zhou, Y., 2015. Estimation of human-induced changes in terrestrial water storage through integration of GRACE satellite detection and hydrological modeling: A case study of the Yangtze River basin. Water Resour. Res. 51 (10), 8494-8516.

Humphrey, V., Gudmundsson, L., 2019. GRACE-REC: a reconstruction of climate-driven water storage changes over the last century. Earth Syst. Sci. Data 11 (3), 1153-1170.

Khan, M.S., Liaqat, U.W., Baik, J., Choi, M., 2018. Stand-alone uncertainty characterization of GLEAM, GLDAS and MOD16 evapotranspiration products using an extended triple collocation approach. Agric. For. Meteorol. 252, 256-268.

Liu, B., Zou, X., Yi, S., Sneeuw, N., Cai, J., Li, J., 2021. Identifying and separating climate- and human-driven water storage anomalies using GRACE satellite data. Remote Sens. Environ. 263, 112559.

Liu, Y., Zhangg, Y., Shan, N., Zhang, Z., Wei, Z., 2022. Global assessment of partitioning transpiration from evapotranspiration based on satellite solar-induced chlorophyll fluorescence data. J. Hydrol. 612, 128044.

Martens, B., Gonzalez Miralles, D., Lievens, H., Van Der Schalie, R., De Jeu, R. A., Fernández-Prieto, D. Verhoest, N. 2017. GLEAM v3: satellite-based land evaporation and root-zone soil moisture. Geosci. Model Dev. 10 (5), 1903-1925.

Miralles, D.G., Holmes, T.R.H., de Jeu, R.A.M., Gash, J.H., Meesters, A.G.C.A., Dolman, A.J., 2011. Global land-surface evaporation estimated from satellite-based observations. Hydrol. Earth Syst. Sci. 15, 453-469.

Rehana, S., Naidu, G.S., 2021. Development of hydro-meteorological drought index under climate change - Semi-arid river basin of Peninsular India. J. Hydrol. 594, 125973.

Rodell, M., Houser, P., Jambor, U.E.A., Gottschalck, J., Mitchell, K., Meng, J., Arsenault, K., Cosgrove, B., Radakovich, J., Bosilovich, M.G., Entin, J., Walker, J., Lohmann, D., Toll, D.L., 2004. The global land data assimilation system. BAMS. 85, 381-394.

Rodell, M., Famiglietti, J.S., Wiese, D.N., Reager, J.T., Beaudoing, H.K., Landerer, F.W., Lo, M.H., 2018. Emerging trends in global freshwater availability. Nature 557, 6510659.

Yang, X., Tian, S., You, W., Jiang, Z., 2021. Reconstruction of continuous GRACE/GRACE-FO terrestrial water storage anomalies based on time series decomposition. J. Hydrol. 603, 127018.

Yao, L., Li, Y., Chen, X., 2021. A robust water-food-land nexus optimization model for sustainable agricultural development in the Yangtze River Basin. Agric. Water Manag. 256, 107103.

(8) L191: The number of hidden neurons of the MLP model has been set to five using a trial-and-error method. How has this been done and are the results reproducible? And did the authors obtain the same optimal number of hidden neurons for each of the four basins?

Response: Thanks for your kind reminder. For a MLP model, the number of neurons in the input layer and that in the output layer are determined by the number of inputs (i.e. three) and outputs (i.e. one), respectively. The number of neurons in the hidden layer determines the structure of MLP model and needs to be optimized through model calibration. Trial-and-error has been a widely used method for selecting the best network structure (Adamowski et al., 2012; Maroufpoor et al., 2020; Patault et al., 2021; Rahman et al., 2020; Sahoo et al., 2008; Sattari et al., 2021; Xie et al., 2019; Xie et al., 2021; Zhou et al., 2020; Zhu et al., 2022;) and is therefore applied in this study to determine the optimum number of neuros in the

hidden layer. Furthermore, a rule-of-thumb suggests that the number of hidden neurons should be more than half the number of inputs but never be more than twice as large when developing a MLP model (Berry and Linoff, 2004; Long et al., 2014). Therefore, the optimum number of neurons in the hidden layer can be obtained by testing different alternative values ranging from two to six and selecting the value with a corresponding ANN model that shows the best performance in the validation set after being trained in the training set using the available data. According to the above procedures, we can eventually set the number of hidden neurons to five via a classic trial-and-error process in the YRB. Similarly, the MLP models in other basins are also tested on a ''trial and error'' basis in order to determine the optimum number of neurons in the hidden layer. Theoretically the optimal number of hidden neurons for each of the four basins may vary from region to region. After further comparing the performance of the MLP model with different hidden neurons in other three subbains, there are no obvious improvements in reconstructing TWSA compared to the MLP model with five hidden neuros. For convenience, the most optimal number of hidden neurons is therefore set to five in four basins in this study.

**References**

Adamowski, J., Chan, H., Prasher, S., Ozga-Zielinski, B., Sliusarieva, A., 2012. Comparison of multiple linear and nonlinear regression, autoregressive integrated moving average, artificial neural network, and wavelet artificial neural network methods for urban water demand forecasting in Montreal, Canada. Water Resour. Res. 48, W01528.

Berry, M.J., Linoff, G.S., 2004. Data mining techniques: For marketing, sales, and customer relationship management. Wiley Computer Publishing.

Long, D., Shen, Y., Sun, A., Hong, Y., Longuevergne, L., Yang, Y., Li, B., Chen, L., 2014. Drought and flood monitoring for a large karst plateau in Southwest China using extended GRACE data. Remote Sens. Environ. 155, 145-160.

Maroufpoor, S., Bozorg-Haddad, O., Eisa, M., 2020. Reference evapotranspiration estimating based on optimal input combination and hybrid artificial intelligent model: Hybridization of artificial neural network with grey wolf optimizer algorithm. J. Hydrol. 588, 125060.

Patault, E., landemaine, V., Ledun, J., Soulignac, A., Fournier, M., Ouvry, J., Cerdan, O., Laignel, B., 2021. Simulating sediment discharge at water treatment plants under different land use scenarios using cascade modelling with an expert-based erosion-runoff model and a deep neural network. Hydrol. Earth Syst. Sci. 25, 6223-6238.

Rahman, S.A., Chakrabarty, D., 2020. Sediment transport modelling in an alluvial river with artificial neural network. J. Hydrol. 588, 125056.

Sahoo, G.B., Ray, C., 2008. Microgenetic algorithms and artificial neural networks to assess minimum data requirements for prediction of pesticide concentrations in shallow groundwater on a regional scale. Water Resour. Res. 44, W05414.

Sattari, M.T., Apaydin, H., Band, S.S., Mosavi, A., Prasad, R., 2021. Comparative analysis of kernel-based versus ANN and deep learning methods in monthly reference evapotranspiration estimation. Hydrol. Earth Syst. Sci. 25, 603-618.

Xie, J., Xu, Y.P., Gao, C., Xuan, W., Bai, Z., 2019. Total basin discharge from GRACE and Water balance method for the 720 Yarlung Tsangpo River basin, Southwestern China. J. Geophys. Res. Atmos. 124, 7617-7632.

Xie, S., Wu, W., Mooser, S., Wang, Q.J., Nathan, R., Huang, Y., 2021. Artificial neural network based hybrid modeling approach for flood inundation modeling. J. Hydrol. 592, 125605.

Zhou, F., Liu, B., Duan, K., 2020. Coupling wavelet transform and artificial neural network for forecasting estuarine salinity. J. Hydrol. 588, 125127.

Zhu, S., Zecchin, A.G., Maier, H.R., 2022. Use of exploratory fitness landscape metrics to better understand the impact of model structure on the difficulty of calibrating artificial neural network models. J. Hydrol. 612, 128093.

(9) L210-229: Are these formulas new or have these formulas already been described in the literature? In the latter case, this part can be removed from the manuscript.

Response: To better introduce the LSTM model, these formulas were presented in this study. Given that these formulas are described in the related literature cited by this study, they have been removed from the manuscript as you suggested.

(10) L234-240: Why is the regression equation linear? Shouldn't any non-linear transformations be considered or is it reasonable to assume linear relations between the three inputs and TWSA?

Response: The reason why we applied the method of multiple linear regression in this study can be explained as follows. As shown in previous studies, the multiple linear regression model can be an alternative way to derive the relationships between the GRACE/GRACE-FO derived TWSA and its predictors (e.g., in situ river gauging data, air temperature, precipitation, soil moisture, climate indices, and other climate variables) (de Linage et al., 2014; Sun et al., 2020; Humphrey et al., 2017; Humphrey and Gudmundsson, 2019; Jing et al., 2020; Li et al., 2020; Nie et al., 2015; Sohoulande et al., 2020; Yang et al., 2021; Zhang et al., 2022). For example, Sun et al., (2020) used a multiple linear regression model

to learn the relationship between the GRACE TWSA and climate drivers or the past and the present TWSA over the 60 basins. Therefore, the method of multiple linear regression was applied as an alternative way (or benchmark model) to analyze the relationships between the GRACE/GRACE-FO derived TWSA and its predictors in this study. It should be noted that although these multiple linear regression models are relatively simple and easy to implement, they are typically reliable and useful only in some specific regions where linear relationships between TWSA and independent variables existed (Yang et al., 2021). As shown in previous studies, changes in TWSA can also be affected by climate through a nonlinear relationship for most regions (de Linage et al., 2014; Sun et al., 2020; Humphrey et al., 2017; Humphrey and Gudmundsson, 2019; Jing et al., 2020). Therefore, another two models, namely the MLP model and the LSTM model, were applied in this study to further analyze the nonlinear relationships between GRACE/GRACE-FO derived TWSA and other climatic variables in the YRB and its sub-basins. After comparing the performances of each model, the specific model which shows the best performance in simulating TWSA can be identified.

**References**

de Linage, C., Famiglietti, J.S., Randerson, J.T., 2014. Statistical prediction of terrestrial water storage changes in the Amazon Basin using tropical Pacific and North Atlantic sea surface temperature anomalies. Hydrol. Earth Syst. Sci. 18 (6), 2089-2102.

Sun, Z., Long, D., Yang, W., Li, X., Pan, Y., 2020. Reconstruction of GRACE data on changes in total water storage over the global land surface and 60 basins. Water Resour. Res. 56, e2019WR026250.

Humphrey, V., Gudmundsson, L., Seneviratne, S.I., 2017. A global reconstruction of climate-driven subdecadal water storage variability. Geophys. Res. Lett. 44 (5), 2300-2309.

Humphrey, V., Gudmundsson, L., 2019. GRACE-REC: a reconstruction of climate-driven water storage changes over the last century. Earth Syst. Sci. Data 11 (3), 1153-1170.

Jing, W., Di, L., Zhao, X., Yao, L., Xia, X., Liu, Y., Yang, J.i., Li, Y., Zhou, C., 2020. A data-driven approach to generate past GRACE-like terrestrial water storage solution by calibrating the land surface model simulations. Adv. Water Resour. Res. 143, 103683.

Li, F., Kusche, J., Rietbroek, R., Wang, Z., Forootan, E., Schulze, K., Lück, C., 2020. Comparison of data-driven techniques to reconstruct (1992-2002) and predict (2017-2018) GRACE-like gridded total water storage changes using climate inputs. Water Resour. Res. 56, e2019WR026551.

Nie, N., Zhang, W., Zhang, Z., Guo, H., Ishwaran, N., 2015. Reconstructed terrestrial water storage change (ΔTWS) from 1948 to 2012 over the Amazon Basin with the Latest GRACE and GLDAS Products. Water Resour. Manage. 30 (1), 279-294.

Sohoulande, C.D.D., Martin, J., Szogi, A., Stone, K., 2020. Climate-driven prediction of land water storage anomalies: an outlook for water resources monitoring across the conterminous United States. J. Hydrol. 588, 125053.

Yang, X., Tian, S., You, W., Jiang, Z., 2021. Reconstruction of continuous GRACE/GRACE-FO terrestrial water storage anomalies based on time series decomposition. J. Hydrol. 603, 127018.

Zhang, B., Yao, Y., He, Y., 2022. Bridging the data gap between GRACE and GRACE-FO using artificial neural network in Greenland. J. Hydrol. 608, 127614.

(11) L243-253: Several issues regarding the flood event selection are not clear. Why were five-day windows selected (step 1)? I can imagine that different windows need to be used for basins of different sizes. What is the background of the factor 1.11 (step 2)? How were the runoff events identified, i.e. how reproducible are the results (step 4)?

Response: Thanks for the valuable comments. As documented in previous studies (Fischer et al., 2021; Giani et al., 2022; Lu et al., 2020; Tarasova et al., 2018; Winter et al., 2021, 2022), this method to select flood event has been widely applied in many different basins over the world because of its advantages in capturing flood events. For example, Lu et al. (2020) applied this nonparametric algorithm to identify runoff events occurred in the Southeastern Coastal Region of China, whose event characteristics are similar to the YRB, with the approach of Tarasova et al. (2018). Considering that it has been successfully applied in many basins over the world including some basins whose event characteristics are similar to the YRB, this nonparametric algorithm suggested by Tarasova et al. (2018) is therefore adopted to identify runoff events in the YRB. The main procedure of this algorithm has been introduced in Section 4.4. Furthermore, as described in Line 228 to Line 230, we also added some necessary sentences in order to explain the reason why we choose the method of flood event selection in this study. According to the study of Tarasova et al. (2018), five-day windows and the factor of 1.11 are selected based on the findings shown in Institute of Hydrology (1980) and World Meteorological Organization (2008) because it proved to be superior in identifying the starting point of potential runoff events (i.e., troughs) and performed consistently in a wide range of catchments, which is beyond the main topic of this study. Nevertheless, much more efforts will be made to analyze the effects of different windows on identify runoff events for basins of different sizes in our future work. Before we start the Step 4, digital filters are previously used to separate the base flow from the total streamflow due to its applicability to large data sets. In Step 4, streamflow time series are screened to identify runoff events after the separation of base flow, which are characterized by their peak, beginning, and end points. As described in Section 4.4, a typical runoff event can be characterized by three main components, namely peak, beginning, and end points. A peak refers

to the maximum of streamflow for a specific period. The beginning point refers to the closest point in time when total runoff is equal to base flow before the peak. Similarly, the end point denotes the closest point in time when total runoff is equal to base flow after the peak. More detailed information about this step can also refer to the study of Tarasova et al. (2018).

"A nonparametric algorithm suggested by Tarasova et al. (2018) is adopted to identify runoff events in this study, which has been widely applied in many different basins over the world because of its advantages in identifying flood events (Fischer et al., 2021; Giani et al., 2022; Lu et al., 2020; Winter et al., 2022)."

**References**

Fischer, S., Schumann, A., Bühler, P., 2021. A statistics-based automated flood event separation. J. Hydrol. X, 10, 100070.

Giani, G., Tarasova, L., Woods, R.A., Rico-Ramirez, M.A., 2022. An objective time-series-analysis method for rainfall-runoff event identification. Water Resour. Res. 58, e2021WR031283.

Institute of Hydrology., 1980. Low flow studies, Report 1, Wallingford, UK.

Lu, W., Lei, H., Yang, W., Yang, J., Yang, D., 2020. Comparison of Floods Driven by Tropical Cyclones and Monsoons in the Southeastern Coastal Region of China. J. Hydrometeorol. 21(7), 1589-1603.

Tarasova, L., Basso, S., Zink, M., Merz, R., 2018. Exploring controls on rainfall-runoff events: 1. Time series-based event separation and temporal dynamics of event runoff response in Germany. Water Resour. Res. 54, 7711-7732.

Winter, C., Lutz, S.R., Musolff, A., Kumar, R., Weber, M., Fleckenstein, J.H., 2021. Disentangling the impact of catchment heterogeneity on nitrate export dynamics from event to longterm time scales. Water Resour. Res. 57, e2020WR027992.

Winter, C., Tarasova, L., Lutz, S.R., Musolff, A., Kumar, R., Fleckenstein, J.H., 2022. Explaining the Variability in High-Frequency Nitrate Export Patterns Using Long-Term Hydrological Event Classification. Water Resour. Res. 58, e2021WR030938.

World Meteorological Organization., 2008. Manual on low-flow estimation and prediction. Operational Hydrology Report No.15.

(12) L276: It would be good to also describe the spatial characteristics (e.g. different basins, averaging) of the analyses.

Response: Thanks for your valuable suggestions. As described in Line 262 to Line 266, we have added some sentences to describe the spatial characteristics (e.g. different basins, averaging) of the analyses based on your suggestions.

"Monthly TWSA estimates during the extreme flood events occurred in the YRB can be reconstructed at regional scales based on the above three different learning-based models, namely the MLP model, the LSTM model and the MLR model. Meanwhile, these three models are further validated in four different basins covering from the upstream to downstream of the Yangtze River in order to better evaluate their applications. More detailed information about all these four different basins can also refer to Table S1."

**Table S1. Long-term hydro-meteorological characteristics of different basin during 2003-2020.**

| Catchment | Gauge station | Area (10$^4$ km$^2$) | Precipitation (mm/yr) | Temperature (°C/yr) | Streamflow (10 m$^8$/yr) | Elevation (m) |
|---|---|---|---|---|---|---|
| SYRB | Shigu | 20.8 | 525 | -0.6 | 437 | 4526 |
| UYRB | Yichang | 99.3 | 843 | 8.1 | 4186 | 2676 |
| UMYRB | Hankou | 140.6 | 973 | 11.3 | 6929 | 1911 |
| YRB | Datong | 181.1 | 1094 | 12.3 | 8781 | 1621 |

Note: SYRB = Source regions of the Yangtze River Basin; UYRB = Upper regions of the Yangtze River Basin; UMYRB = Upper and the Middle regions of the Yangtze River Basin; YRB = Yangtze River Basin.

(13) L308-309: Why is TWSA consistent with temperature? What is the physical mechanism causing a high (positive) correlation between these variables?

Response: The physical mechanism causing a high (positive) correlation between the temperature and TWSA can be explained as follows. Temperature, representing the available energy over regions, is an important proxy for evapotranspiration (ET) because it can indirectly reflect evaporation from water bodies and soil layers in the humid region and therefore be linked to surface water and groundwater, both of which are essential components of TWSA. In other words, temperature plays a critical role in variation of ET while the latter is closely correlated with changes in TWSA (i.e. TWSC) according to the water balance method (Long, et al., 2014; Lv et al., 2017; Xie et al., 2019) (Eq. (1)).

$$TWSC = P - R - ET \tag{1}$$

where TWSC represents changes in TWSA for regions, P represents precipitation, R and ET represents the outflow and evapotranspiration for a specific region of interest.

Therefore, temperature has been viewed as one of the most important factors that can influence the variation of TWSA and shows a high correlation with TWSA in many studies (Humphrey et al., 2017; Long, et al., 2014; Li et al., 2020; Trautmann et al., 2018). For example, de Linage et al., (2014) examined how sea surface temperature anomalies influence interannual variability of TWSA in different regions within the Amazon Basin and found that interannual TWSA in tropical South America was significantly influenced by variations in sea surface temperature anomalies. Deng et al., (2017) investigated the effects of climate change on TWSA in the Central Asia and found that there was a significant positive correlation between TWSA and temperature in the southwestern and southeastern regions of the Central Asia.

In addition, changes in temperature can also influence the variation of glaciers or snow in the upper stream of the study area, which are components of regional TWSA. Given that the above reasons, temperature has been usually selected as an essential variable when reconstructing the time series of TWSA for regions.

**References**

de Linage, C., Famiglietti, J.S., Randerson, J.T., 2014. Statistical prediction of terrestrial water storage changes in the Amazon Basin using tropical Pacific and North Atlantic sea surface temperature anomalies. Hydrol. Earth Syst. Sci. 18 (6), 2089-2102.

Deng H., Chen, Y., 2017. Influences of recent climate change and human activities on water storage variations in Central Asia. J. Hydrol. 544, 46-57.

Humphrey, V., Gudmundsson, L., Seneviratne, S.I., 2017. A global reconstruction of climate-driven subdecadal water storage variability. Geophys. Res. Lett. 44 (5), 2300-2309.

Li, F., Kusche, J., Rietbroek, R., Wang, Z., Forootan, E., Schulze, K., Lück, C., 2020. Comparison of data-driven techniques to reconstruct (1992-2002) and predict (2017-2018) GRACE-like gridded total water storage changes using climate inputs. Water Resour. Res. 56, e2019WR026551.

Long, D., Shen, Y., Sun, A., Hong, Y., Longuevergne, L., Yang, Y., et al. (2014). Drought and flood monitoring for a large karst plateau in Southwest China using extended GRACE data. Remote Sensing of Environment, 155, 145-160.

Lv, M., Ma, Z., Yuan, X., Lv, M., Li, M., & Zheng, Z. (2017). Water budget closure based on grace measurements and reconstructed evapotranspiration using GLDAS and water use data for two large densely-populated mid-latitude basins. Journal of Hydrology, 168, 177-193.

Trautmann, T., Koirala, S., Carvalhais, N., Eicker, A., Fink, M., Niemann, C., Jung, M., 2018. Understanding terrestrial water storage variations in northern latitudes across scales. Hydrol. Earth Syst. Sci. 22 (7), 4061-4082.

Xie, J., Xu, Y.P., Gao, C., Xuan, W., Bai, Z., 2019. Total basin discharge from GRACE and Water balance method for the Yarlung Tsangpo River basin, Southwestern China. J. Geophys. Res. Atmos. 124, 7617-7632.

(14) L362-363: Are the relationships between TWSA and hydroclimatic factors indeed complicated? Figure 4 suggests that these relations are rather straightforward as also confirmed by the authors when using linear correlations for the relations.

Response: As shown in Figure 4, regional TWSA indeed shows a close correlation with precipitation and temperature because precipitation is the main input flux of terrestrial water while temperature represents available energy over regions, both of which play key roles in the regional hydrological cycles. Ignoring anthropogenic influences, all these hydro-climatic factors theoretically dominate changes in TWSA for regions, which is in line with previous studies (Bai et al., 2022; Humphrey et al., 2019; Syed et al., 2008; Xie et al., 2019; Yang et al., 2021; Zhang et al., 2019).

In addition, all of these variables present significantly seasonal variations as shown in Figure 4, which can also explain the close correlation between regional TWSA with precipitation and temperature to some degree. Nevertheless, the results shown in Figure 5 indicate that the MLP model can simulate TWSA better than the multiple linear regression (MLR) model for all basins, indicating that the relationships between TWSA and hydro-climatic factors might not be simple linear correlations. Therefore, two another learning-based models, namely MLP model and the LSTM model, are applied to analyze the relationships between TWSA and hydro-climatic factors besides the MLR model considering its further applications in other basins. The reason why we apply the method of MLR to reconstruct TWSA and more explanations about the relationships between TWSA and hydro-climatic factors can also be found in our responses to comment #10 and comment #13.

**References**

Bai, H., Ming, Z., Zhong, Y., Zhong, M., Kong, D., Ji, B., 2022. Evaluation of evapotranspiration for exorheic basins in China using an improved estimate of terrestrial water storage change. J. Hydrol. 610, 127885.

Humphrey, V., Gudmundsson, L., 2019. GRACE-REC: a reconstruction of climate-driven water storage changes over the last century. Earth Syst. Sci. Data 11 (3), 1153-1170.

Syed, T.H., Famiglietti, J.S., Rodell, M., Chen, J., Wilson, C.R., 2008. Analysis of terrestrial water storage changes from GRACE and GLDAS. Water Resour. Res. 44 (2), 339-356.

515   Xie, J., Xu, Y.P., Wang, Y., Gu, H., Wang, F., Pan, S., 2019. Influences of climatic variability and human activities on terrestrial water storage variations across the Yellow River basin in the recent decade. J. Hydrol. 579, 124218.

Yang, X., Tian, S., You, W., Jiang, Z., 2021. Reconstruction of continuous GRACE/GRACE-FO terrestrial water storage anomalies based on time series decomposition. J. Hydrol. 603, 127018.

520   Zhang, Y., He, B., Guo, L., Liu, J., Xie, X., 2019. The relative contributions of precipitation, evapotranspiration, and runoff to terrestrial water storage changes across 168 river basins. J. Hydrol. 579, 124194.

(15) L368-369: Is the limited availability of training data the main reason for the moderate performance of the LSTM model? In this study, it seems that sufficient data are available for training and validation.
525   Another factor which might (partly) explain the moderate performance of this model compared to the MLP model is the possibly limited role of the memory function in the LSTM model, since relations between inputs and the output are quite direct without much memory effects.

Response: Thanks for your valuable suggestions. As described in Line 356 to Line 359, we have added some sentences and references to better explain the moderate performance of the LSTM model in
530   reconstructing TWSA based on your suggestions. Furthermore, we mainly focus on the relationships between TWSA and hydro-climatic factors in this study due to the limitation of data available. Therefore, the LSTM method for temporally downscaling monthly TWSA derived from GRACE/GRACE-FO satellite data should be further assessed by incorporating with more related factors in the future. More detailed exaltations about the relationships between TWSA and hydro-climatic factors can also be found
535   in our response to Comment #14.

"In addition, the moderate performance of LSTM model in reconstructing TWSA compared to the MLP model can be partly attributed to the possibly limited role of the memory function in the LSTM model (Wei et al., 2021; Yin et al., 2022), since relations between inputs and the output of this model (shown in Figure 4) are pretty direct without much memory effects."

540   **References**

Wei, L., Jiang, S., Ren, L., Tan, H., Ta, W., Liu, Y., Yang, X., Zhang, L., Duan, Z., 2021. Spatiotemporal changes of terrestrial water storage and possible causes in the closed Qaidam Basin, China using GRACE and GRACE Follow-On data. J. Hydrol. 598, 126274.

Yin, H., Wang, F., Zhang, X., Zhang, Y., Chen, J., Xia, R., Jin, J., 2022. Rainfall-runoff modeling using long short-term memory based step-sequence framework. J. Hydrol. 610, 127901.

(16) L374-378: An important assumption in this study is the independency of the input-output relations as determined by the data-based models on time scales, i.e. the relations found from monthly data are also assumed to be valid for daily data. The authors use as argument that 'the same scaling properties have been commonly assumed for baseline and future periods in temporal downscaling'. However, the situation/ conditions mentioned in this argument (translation from one period to another) is different from the situation relevant in this study (translation from one (coarse) time scale to another (fine) time scale). Relations between (hydro)climatic variables and (other) hydrological variables are very different at different time scales. For instance, rainfall-runoff relations at hourly or daily time steps usually are highly non-linear, where relations at monthly or annual time scales are more or almost linear. Hence, it is doubtful whether the input-output relations at monthly time scales established by the three data driven models (in particular the MLP model) can be used for daily time steps as well. The authors should try to use some independent data sources (e.g. groundwater level measurements) at sub-monthly (preferably daily) scales to test the downscaled relations. Without such a validation/ testing it will be hard to assess the reliability of the results.

Response: Thanks for your valuable suggestions. We greatly agree with you that it is meaningful to use more independent data sources (e.g. groundwater level measurements) at sub-monthly (preferably daily) scales to test the downscaled relations. Therefore, we further compare the temporally downscaled TWSA estimates with the daily TWSA estimates derived from the ITSG-GRACE2018, which is a daily TWSA dataset, considering that other independent data sources (e.g. groundwater level measurements) are not available indeed in this study. ITSG-GRACE2018 is derived from GRACE Level-1B data and generated at the Institute of Geodesy of Graz University of Technology (ITSG) (Kvas et al., 2019), which has been proved to be effective in tracking TWSA variation at a daily scale (Bai et al., 2022; Eicker et al., 2020; Xiong et al., 2022; Gouweleeuw et al., 2018; G ür et al.,2018). For further detailed descriptions about ITSG-GRACE2018, the reader is referred to the website of https://www.tugraz.at/institute/ifg/downloads/ gravity-field-models/itsg-grace2018/. As shown in Figure R2, daily TWSA estimates based on the MLP model are significantly consistent with that derived from the ITSG-GRACE2018 particularly in the YRB. The highest coefficient of determination between daily TWSA (mm) derived from ITSG-GRACE 2018 and that simulated by the MLP model is observed in the YRB with $R^2 = 0.72$ (p < 0.01), indicating that the method applied in this study is effective and acceptable in obtaining a daily time series of TWSA when independent data sources such as groundwater level measurements are not available. Furthermore, some sentences are also added in Section 6.3, emphasizing that more efforts should be made to validate

the reliability of downscaled relations in our next study when more independent data sources are available based on your suggestions.

580 "Additionally, more efforts should be made to further validate the reliability of temporally downscaled relations proposed in this study when more independent data sources (e.g. groundwater level measurements) are available in YRB."

[Figure]

**Figure R2: Comparison between daily TWSA (mm) derived from ITSG-GRACE 2018 and that simulated by the MLP model for (a) the SYRB, (b) the UYRB, (c) the UMYRB and (d) the YRB respectively. TWSA= Terrestrial water**
585 **storage anomaly; ITSG = Institute of Geodesy of Graz University of Technology; GRACE = Gravity Recovery and Climate Experiment; MLP = Multi-layer perceptron; SYRB = Source regions of Yangtze River Basin; UYRB = Upper regions of Yangtze River Basin; UMYRB = Upper and middle regions of Yangtze River Basin; YRB = Yangtze River Basin. Note that the TWSA data derived from ITSG-GRACE2018 only covered the period between 2003 and 2016.**
**References**

Bai, H., Ming, Z., Zhong, Y., Zhong, M., Kong, D., Ji, B., 2022. Evaluation of evapotranspiration for exorheic basins in China using an improved estimate of terrestrial water storage change. J. Hydrol. 610, 127885.

Eicker, A., Jensen, L., Wöhnke, V., Doslaw, H., Kvas, A., Mayer-Gürr, T., Dill, R., 2020. Daily GRACE satellite data evaluate short-term hydro-meteorological fluxes from global atmospheric reanalyses. Sci. Rep. 10, 4504

Gouweleeuw, B.T., Kvas, A., Gruber, C., Gain, A.K., Mayer-Guerr, T., Flechtner, F., Guentner, A., 2018. Daily GRACE gravity field solutions track major flood events in the Ganges-Brahmaputra Delta. Hydrol. Earth Syst. Sci. 22, 2867-2880.

Gürr, M., Saniya, B., Matthias, E., Andreas, K., Beate, K., Sebastian, S., Norbert, Z., 2018. ITSG Grace2018-monthly, daily and static gravity field solutions from GRACE. GFZ Data Services.

Kvas, A., Behzadpour, S., Ellmer, M., Klinger, B., Strasser, S., Zehentner, N., Mayer-Gürr, T., 2019. ITSG-Grace2018: overview and evaluation of a new GRACE-only gravity field time series. J. Geophys. Res. Solid Earth 124 (8), 9332-9344.

Xiong, J., Guo, S., Yin, J., Ning, Z., Zneg, Z., Wang, R., 2022. Projected changes in terrestrial water storage and associated flood potential across the Yangtze River basin. Sci. Total Environ. 817, 152998.

(17) L379-381: The relations determined at a monthly scale are used with daily inputs as well. I would not call this downscaling as the same relations are used for different time steps.

Response: Thanks for your valuable suggestions. We have further validated the reliability of temporally downscaled TWSA by comparing with the results derived from another dataset as described in detail in our response to Comment #17. Meanwhile, some sentences are also added in Section 6.3, emphasizing that more efforts should be made to validate the reliability of downscaled relations in our next study when more independent data sources are available. Furthermore, we have revised the title of this manuscript with "Monitoring the extreme flood events in the Yangtze River Basin based on GRACE/GRACE-FO satellite data" based on your suggestions.

"Additionally, more efforts should be made to further validate the reliability of temporally downscaled relations proposed in this study when more independent data sources (e.g. groundwater level measurements) are available in YRB."

(18) L385-387: The results in Figure 6 show that the daily TWSA estimates do not capture the minimum monthly TWSA observations in all basins for several years. That is opposite to what the authors mention in this sentence. It would be good to also compare daily and monthly TWSA estimates in addition to the comparison between daily TWSA estimates and monthly TWSA observations. This might partly explain this behavior.

Response: Thanks for your valuable suggestions. We have revised Figure 6 and further added the monthly TWSA estimates estimated by the MLP model based on your suggestions. As shown in Figure 6, daily and monthly TWSA estimates estimated by the MLP model and monthly estimates derived from GRACE/GRACE-FO satellite data are presented in this figure.

[Figure]

**Figure 6: Daily (TWSA-MLP-day) and monthly (TWSA-MLP-month) time series of TWSA simulated by the MLP model for (a) the SYRB, (b) the UYRB, (c) the UMYRB and (d) the YRB respectively during 2003-2020. Note that monthly TWSA estimates derived from GRACE/GRACE-FO satellite data (TWSA-GRACE-month) shown in this figure are detrended because hydro-climatic factors may not fully simulate their long-term trends. TWSA = Terrestrial water storage anomaly; MLP = Multi-layer perceptron neural network; SYRB = Source regions of Yangtze River Basin; UYRB = Upper regions of Yangtze River Basin; UMYRB = Upper and middle regions of Yangtze River Basin; YRB = Yangtze River Basin.**

(19) L401-402: It is doubtful whether soil moisture is the dominant component of TWSA. Most water is stored as (saturated) groundwater or surface water and the soil moisture storage is limited compared to these two storage components. Also changes in groundwater and surface water storage can generally be much larger than changes in soil moisture storage.

Response: Thanks for your valuable suggestions. As described in Line 401 to Line 403, we have carefully checked and revised this sentence to avoid ambiguity. In addition, we also added some necessary references to make the explanation clear.

"This is partly because high antecedent soil moisture, which is a component of TWSA, has been identified as an important driver of flood events for regions (Fatolazadeh et al., 2022; Jing et al., 2020; Reager et al., 2014; Wasko et al., 2019)."

**References**

Fatolazadeh, F., Goïa, Kalifa., 2022. Reconstructing groundwater storage variations from GRACE observations using a new Gaussian-Han-Fan (GHF) smoothing approach. J. Hydrol. 604, 127234.

Jing, W., Di, L., Zhao, X., Yao, L., Xia, X., Liu, Y., Yang, J.i., Li, Y., Zhou, C., 2020. A data-driven approach to generate past GRACE-like terrestrial water storage solution by calibrating the land surface model simulations. Adv. Water Resour. Res. 143, 103683.

Reager, J.T., Thomas, B.F., Famiglietti, J.S., 2014. River basin flood potential inferred using GRACE gravity observations at 665 several months lead time. Nat. Geosci. 7 (8), 588-592.

Wasko, C., Natthan, R., 2019. Influence of changes in rainfall and soil moisture on trends in flooding. J. Hydrol. 575, 432-441.

(20) L444-446: Do the lag times vary with the size of the basins? I can imagine that larger basins (e.g. the entire YRB) will have larger lag times than smaller basins.

Response: Yes. The lag times may vary with the size of the basins as you mentioned. More specific, larger basins (e.g. the entire YRB) generally have larger lag times than smaller basins for extreme flood events. Above all, the comparison analysis indicates that the flood events can be monitored by the proposed NDFPI earlier than traditional streamflow observations for all basins, which is very vital for flood forecasts and warning across this region.

(21) L478-485: The advantages of the proposed method and possible operationalisation need some nuance. Besides the considerations mentioned under specific comment 16, it should be emphasized that operationalisation would mean application of the MLP model at a daily scale using daily temperature, precipitation and soil moisture storage as inputs. In fact, the TWSA observations are not directly used for early flood detection, but only indirectly for establishing relations at a monthly scale which are used at a daily scale. As such, the early flood detection is still mainly based on in situ data and will be hardly applicable to poorly or ungauged areas. Hence, the authors are encouraged to investigate whether observed TWSA estimates at a monthly time scale can be downscaled to a daily time scale without using in situ data or at least without using these data in an operational context.

Response: We agree with the reviewer that it is very meaningful to investigate whether observed TWSA estimates at a monthly time scale can be downscaled to a daily time scale without using in situ data. Although the flood potential index proposed in this study is estimated by jointly using the GRACE satellite data and meteorological data, it can provide a simple but useful framework for monitoring the flood events and its hydrological impacts at finer temporal scales for study regions. Furthermore, the methods and conclusions shown in this study can provide broader implications for flood monitoring in ungauged or poorly gauged basins. For example, advances in satellite remote sensing have made remote sensing a promising approach to capture various hydrological variables (e.g. precipitation, temperature and soil moisture), since they can substantially reduce the limitations of traditional ground-based observations. This is extremely useful and important in hydrological research and applications particularly in ungauged or poorly gauged basins. Recent studies (Chen et al., 2017; Famiglietti et al., 2015; Gao et al., 2017; Li et al., 2019; Samain et al., 2012) revealed that remote sensing data could be used as a valuable alternative to in situ observations as forcing data particularly over ungauged or poorly gauged basins where ground-based observations are extremely limited, because their resolutions and availability were not significantly influenced by climatic conditions or terrain. Therefore, we can calculate the flood potential index proposed in this study (i.e. NDFPI) by jointly using remote sensing-based precipitation, temperature and soil moisture estimates combined with GRACE/GRACE-FO satellite data, which can further provide the potential for flooding in ungauged or poorly gauged basins during the study period. More detailed description about multiple remote sensing data except for GRACE/GRACE-FO satellite data can also refer to Table R2. In brief, much further work, including monitoring and predictions of extreme floods using pure remotely sensed data products, is urgently needed as you stated in this question. As described in Line 543 to Line 556, we have added some sentences to better show the important implications provided by this study.

Table R2. Detailed description about multiple remote sensing data.

| Data types | Data sources | Spatial Resolution | Temporal Resolution | Period |
|---|---|---|---|---|
| Temperature | MODIS | day | 1 km | 2000-2022 |
| Precipitation | TRMM-3B42 | day | 0.25° | 1998-2020 |
| | GPM | day | 0.25° | 2000-2022 |
| | CMORPH_CDR | day | 0.25° | 1998-2021 |
| Soil moisture | SMAP | day | 1 km | 2015-2022 |
| | SMOS | day | 25 km | 2010-2022 |

Note: MODIS = Moderate-resolution Imaging Spectroradiometer; SMOS = Soil Moisture and Ocean Salinity; SMAP = Soil Moisture Active and Passive; TRMM = Tropical Rainfall Measuring Mission; CMORPH-CDR = Climate Prediction Center morphing method Climate Data Record; GPM = Global Precipitation Measurement.

"Furthermore, this study can also provide broader implications for flood monitoring in ungauged or poorly gauged basins. For example, advances in satellite remote sensing have made remote sensing a

promising approach to capture various hydrological variables (e.g. precipitation, temperature and soil moisture), since they can substantially reduce the limitations of traditional ground‑based observations. This is extremely useful and important in hydrological research and applications particularly in ungauged or poorly gauged basins. Therefore, we can calculate the flood potential index proposed in this study (i.e. NDFPI) by jointly using remote sensing-based precipitation, temperature and soil moisture estimates combined with GRACE/GRACE-FO satellite data, which can further provide the potential of remote sensing data for flooding in ungauged or poorly gauged basins."

**References**

Chen, X., Long, D., Hong, Y., Zeng, C., Yan, D., 2017. Improved modeling of snow and glacier melting by a progressive two-stage calibration strategy with GRACE and multisource data: How snow and glacier meltwater contributes to the runoff of the Upper Brahmaputra River basin? Water Resour. Res. 53, 2431-2466.

Famiglietti, J.S., Cazenave, A., Eicker, A., Reager, J.T., Rodell, M., Velicogna, I., 2015. Satellites provide the big picture. Science 349 (6249), 684-685.

Gao, Z., Long, D., Tang, G., Zeng, C., Huang, J., Hong, Y., 2017. Assessing the potential of satellite-based precipitation estimates for flood frequency analysis in ungauged or poorly gauged tributaries of China's Yangtze River basin. J. Hydrol. 550, 478-496.

Li, X., Long, D., Han, Z., Scanlon, B. R., Sun, Z., Han, P., Hou, A., 2019. Evapotranspiration estimation for Tibetan Plateau headwaters using conjoint terrestrial and atmospheric water balances and multisource remote sensing. Water Resour. Res. 55, 8608-8630.

Samain, B., Simons, G. W. H., Voogt, M. P., Defloor, W., Bink, N. J., Pauwels, V. R. N., 2012. Consistency between hydrological model, large aperture scintillometer and remote sensing based evapotranspiration estimates for a heterogeneous catchment. Hydrol. Earth Syst. Sci. 16(7), 2095-2107.

(22) L517-518: Only water consumption data or also for instance data on reservoir operation?

Response: Thanks for your kind reminder. In addition to water consumption, more detailed statistics related to human use such as reservoir operation and inter-basin water diversion projects should be further considered in our next study when reconstructing the series of regional TWSA across the YRB. Therefore, we have rewritten this sentence to show all factors that should be taken into consideration as you suggested (Line 513 to Line 515).

730       "In future, more attentions should be paid to reconstruct the series of regional TWSA under climatic variability when more detailed statistics related to human use such as water consumption, reservoir operation and inter-basin water diversion projects are available."

**Technical corrections**

L8: 'monitor flood' instead of 'monitor the floods'.

735 Response: As described in Line 8, we have changed "monitor the floods" with "monitor flood" after carefully checking this sentence.

      "Gravity Recovery and Climate Experiment (GRACE) and its successor GRACE Follow-on (GRACE-FO) satellite provide terrestrial water storage anomaly (TWSA) estimates globally that can be used to **monitor flood** in various regions at monthly intervals."

740 L20: 'satellite data' instead of 'satellites data'.

Response: As described in Line 20, we have changed "satellites data" with "satellite data". In addition, we have carefully checked the other related terms through this manuscript and made proper changes.

      "All these findings can provide new opportunities for applying GRACE/GRACE-FO **satellite data** to investigations of sub-monthly signals and have important implications for flood hazard prevention and 745 mitigation in the study region."

L25: 'floods' instead of 'flood'.

Response: As described in Line 25, we have changed "flood" with "floods".

      "According to the report published by the United Nations Office for Disaster Risk Reduction (UNDRR), the total economic loss induced by **floods** is up to \$651 billion (USD) worldwide from 2000 750 to 2019 (https://www.undrr.org/publication/human-cost-disasters-overview-last-20-years-2000-2019)."

L33: 'cycle' instead of 'cycles'.

Response: As described in Line 60, we have changed "cycles" with "cycle".

      "During the past decades, the increasingly intensified human activities and climate change have substantially changed the hydrological **cycle** in the YRB and thus accelerated the variation of flood 755 characteristics in this region (Fang et al., 2012; Wang et al., 2011)."

L35: 'in recent decades' instead of 'in the recent decades'.

Response: As described in Line 62, we have changed "in the recent decades" with "in recent decades" based on your suggestions.

"It has been found that both the frequency and severity of extreme flood events generally showed an upward trend in the YRB **in recent decades** owing to substantial changes in climate, infrastructure and land use (Huang et al., 2015; Liu et al., 2019; Yang et al., 2021; Zhang et al., 2008)."

L48: 'based on a copula function' instead of 'based on the copula function'.

Response: As described in Line 39, we have changed "based on the copula function" with "based on a copula function" based on your suggestions.

"Xiong et al. (2021) developed a novel integrated flood potential index by linking the flood potential index derived from six GRACE products **based on a copula function**, which was further used to identify and characterize the floods with different intensities over the study region."

L57: 'few attention' instead of 'few attentions'.

Response: As described in Line 49, we have changed "attentions" with "attention" based on your suggestions.

"To date, **very few studies have paid attention** to monitor flood events at sub-monthly time scales using GRACE data."

L59: 'using temporally downscaled GRACE data' instead of 'using the temporally downscaled GRACE data'.

Response: As described in Line 51, we have deleted the phrase "using the temporally downscaled GRACE data" based on the suggestions from Reviewer Abhishek Abhi.

L62: 'to monitor extreme flood events' instead of 'to monitor the extreme flood events'. Similar corrections should be made in the remainder of the manuscript. Preferably, a native English speaking person should check and correct the document.

Response: As described in Line 53, we have changed "to monitor the extreme flood events" with "to monitor extreme flood events" based on your suggestions. Furthermore, we have carefully checked the

other terms through this manuscript and tried our best to improve the manuscript both in the language and contents based on your comments and suggestions.

"Therefore, we aim to downscale the TWSA estimates derived from GRACE/GRACE-FO satellite data into daily values and demonstrate its application to **monitor extreme flood events** at sub-monthly time scales for the YRB."

L80: What do the authors mean with 'three-step ladder distribution'? Why three steps and not e.g. two, four or five steps?

Response: To avoid ambiguity, we have deleted the expression of "three-step ladder distribution" shown in this sentence as described in Line 82 to Line 83.

"The terrain of the YRB generally decreases from west to east with altitudes ranging from -142 m to 7143 m above the sea level (shown in Fig. 1)."

L100: 'can be found in Fig 1' instead of 'can refer to Fig. 1'.

Response: As described in Line 103, we have changed "can refer to Fig. 1" with "can be found in Fig 1" based on your suggestions.

"More information about the location and topography of the YRB **can be found** in Fig. 1."

L118: 150 stations for both precipitation and temperature?

Response: Yes. As stated in Section 3.2, daily time series of precipitation and temperature from 2003-2020 can be acquired from all these 150 stations distributed in the YRB (shown in Figure 1).

L142: 'to daily and monthly estimates' instead 'o daily and monthly estimates'.

Response: Thanks for your reminder. As described in Line 148, we have rewritten this sentence based on your Comment #6.

"Furthermore, the temporal resolution of original SMS derived from GLDAS 2.1 Noah land surface model can be decreased from 3-hours to 1-day and 1-month composite respectively, which is consistent with the methods applied in previous studies (Mulder et al., 2015; Mohanasundaram et al., 2021; Syed et al., 2008)."

L153: 'which consists of four steps' instead of 'which is made of four steps'.

Response: As described in Line 155, we have changed "which is made of four steps" with "which consists of four steps" based on your suggestions.

810       "A detailed flow diagram of our study is given in Fig. 2, **which consists of four steps.**"

L158: The role of scenarios is not clear yet.

Response: As described in Line 160 to Line 164, we have rewritten this sentence when describing the role of scenarios in order to make it clear in the manuscript. In addition, we also described in detail the role of three scenarios in Line 266 to Line 269 as you suggested.

815       "Given that different periods of data used for training and validation might influence the performances of each model in simulating TWSA, a total of three scenarios are therefore designed according to the way of dividing training periods and validation periods for each model. After comparing the performances of each model in simulating monthly TWSA estimates under all three scenarios, the calibrated parameter sets of the model with a specific scenario that shows the best performance in
820   simulating monthly TWSA estimates are identified and retained."

       "According to previous findings in Liu et al. (2021), different periods of data used for training (i.e. identification of model parameter sets) and validation can eventually influence the corresponding performances of a specific model when simulating TWSA. Therefore, we design a total of three scenarios according to the way of dividing training periods and validation periods for a specific model."

825  L159: What is meant with 'scaling properties'? This is not clear here.

Response: Thanks for your valuable suggestions. As described in Line 162 to Line 164, we have written this sentence to better explain the specific details of Step 2.

       "After comparing the performances of each model in simulating monthly TWSA estimates under all three scenarios, the calibrated parameter sets of the model with a specific scenario that shows the best
830  performance in simulating monthly TWSA estimates are identified and retained."

L161: 'Step 2' instead of 'Step 3'?

Response: As described in Line 166, we have changed "Step 3" with "Step 2" based on your suggestions.

L200: 'existing' instead of 'existed'.

Response: As described in Line 204, we have changed "existed" with "existing" based on your suggestions.

   "Long short-term memory network (LSTM) is one of the most representative RNNs as it has the fabulous memory ability and can effectively avoid the vanishing gradient problem **existing** in other RNNs (Hochreiter, 1997; Guo et al., 2021)."

L240-242: Shouldn't this sentence be moved to section 4.6?

Response: As described in Line 262 to Line 263, this sentence has been moved to section 4.6 based on your suggestions.

   "Monthly TWSA estimates during the extreme flood events occurred in the YRB can be reconstructed at regional scales based on the above three different learning-based models, namely the MLP model, the LSTM model and the MLR model."

L252-253: Reformulate this sentence.

Response: Thanks for your reminder. As described in Line 237, we have rewritten this sentence as you suggested. More detailed information about the method of identifying runoff events can also refer to our response to Comment #11.

   "Similarly, the end point denotes the closest point in time when total runoff is equal to base flow after the peak."

L270-271: 'the typical difference between the storage change and precipitation'; what do the authors mean with this?

Response: We apologize for such confusion because of the wording. As described in Line 254 to Line 260, we have written this sentence to avoid ambiguity.

   "Finally, we can calculate the normalized daily flood potential index (NDFPI) from the DFPA with the goal of removing the effects of hydrological heterogeneity varying from region to region (Reager et al., 2009), which can be described as follows:

$$NDFPI(t) = \frac{DFPA(t) - DFPA_{min}}{DFPA_{max} - DFPA_{min}},$$ (4)

where $DFPA_{max}$ and $DFPA_{min}$ represent the maximum DFPA and minimum DFPA during the study period respectively. The NDFPI indicates the corresponding probability of flood occurrence with a range from 0 to 1. More flood is likely to occur when the NDFPI is closer to 1 for a specific region."

**References:**

Reager, J.T., Famiglietti, J.S., 2009. Global terrestrial water storage capacity and flood potential using GRACE. Geophys. Res. Lett. 36, L23402.

L280: Remove 'generally'.

Response: As described in Line 268, we have deleted this word based on your suggestions.

"Therefore, we design a total of three scenarios according to the way of dividing training periods and validation periods for a specific model."

L290-295: The subscripts in the formulas are not clear. Subscripts are used for both the time step and observed values, simulated values do not have an additional subscript. For average values an unclear subscript is used as well without having a clear meaning or role. The authors are advised to use subscripts for observed and simulated values (i.e. 'o' and 's') and put the time step between brackets.

Response: Thanks for your valuable suggestions. As described in Line 274 to Line 281, we have revised the subscripts in the formulas based on your suggestions.

"Furthermore, three kinds of statistical measures including root mean square error (RMSE), correlation coefficient (r), and Nash-Sutcliffe efficiency coefficient (NSE) are used in this study as they can jointly measure the matching quality in terms of both magnitude and phase between the simulated and the observed time series. These statistical measures are defined as:

$$RMSE = \sqrt{\frac{\sum_{i=1}^{N}(x_{s,i}-x_{o,i})^2}{N}} \tag{5}$$

$$r = \frac{\sum_{i=1}^{N}(x_{s,i}-\bar{x}_{s,i})(x_{o,i}-\bar{x}_{o,i})}{\sqrt{\sum_{i=1}^{N}(x_{s,i}-\bar{x}_{s,i})^2 \times \sum_{i=1}^{N}(x_{o,i}-\bar{x}_{o,i})^2}} \tag{6}$$

$$NSE = 1 - \frac{\sum_{i=1}^{N}(x_{s,i}-x_{o,i})^2}{\sum_{i=1}^{N}(x_{o,i}-\bar{x}_{o,i})^2} \tag{7}$$

where $x_{s,i}$ and $x_{o,i}$ represent simulated and observed TWSA in month $i$, respectively; $\bar{x}_{s,i}$ and $\bar{x}_{o,i}$ represent the average of simulated and observed TWSA series; $N$ is the total months of observed (or simulated) TWSA available."

L302-303: This sentence includes quite some repetition.

Response: Thanks for your reminder. As described in Line 302, we have deleted this sentence to avoid repetition based on your suggestions.

L306: Is the maximum TWSA value 130.9 mm (this line) or 130.1 mm (line 300)?

Response: Thanks for your reminder. The maximum TWSA value is 130.9 mm after we carefully checking the results shown in Figure 4. Therefore, we have rewritten the sentence shown in Line 286 to keep consistent with the results shown in Line 292.

Line 286: "The results show that monthly TWSA over the YRB has a wide range from -58.0 mm to 130.9 mm during the study period."

Line 292: "Accordingly, TWSA reaches its maximum in July 2020 with an estimate of 130.9 mm during 2003-2020, reflecting the evolution of TWSA in response to heavy rainfall during this period."

L312: 'resulting' instead of 'resulted'.

Response: As described in Line 298, we have changed "resulted" with "resulting" based on your suggestions.

"This phenomenon can be explained by the discrepancies **resulting** from the components of SMSA and TWSA."

L349: 'difference' instead of 'decrease' (two times).

Response: As described in Line 335, we have changed "decrease" with "difference" based on your suggestions.

"…, which is lower than that of 15.1 mm/month for the LSTM-derived TWSA estimates (~39% **difference**) and that of 13.3 mm/month for the MLR-derived TWSA estimates (~22% **difference**)."

L361: 'of' instead of 'higher than' and 'of' instead of 'lower than'?

Response: As described in Line 347, we have changed "higher than" with "of" and changed "lower than" with "of" based on your suggestions.

"Overall, Fig. 5 clearly suggests the MLP model's superior performances in simulating TWSA with an average value of NSE **of** 0.81 and an average value of RMSE **of** 12.8 mm/month during the validation periods for all regions, …"

L368: 'ANN models' instead of 'ANNs models'.

Response: As described in Line 355, we have changed "ANNs models" with "ANN models" based on your suggestions.

"…, which indicates that the LSTM model may not show better performances in simulating time series data than other traditional **ANN models** in some cases especially when limited trained data are available."

L371: 'satellite data' instead of 'satellites data'.

Response: As described in Line 362, we have changed "satellites data" with "satellite data". In addition, we have carefully checked the other related terms through this manuscript and made proper changes.

"5.3 Temporal downscaling of GRACE/GRACE-FO **satellite data**"

L454: 'Discussion' instead of 'Discussions'.

Response: As described in Line 453, we have changed "Discussions" with "Discussion".

L460: Is hysteresis the right term here?

Response: As described in Line 457 to Line 459, we have rewritten this sentence to avoid ambiguity based on your suggestions.

"This is consistent with the results found at the Missouri River basin by Reager et al. (2014). Reager et al. (2014) indicated that regional TWSA may lead river discharge slightly before the flood season, which can provide useful information on the signal of high streamflow in the coming flood season."

L461: What do the authors mean with the term 'predisposition'?

Response: We apologize for such confusion because of the wording. As described in Line 457 to Line 459 we have written this sentence to make the explanation clear.

"This is consistent with the results found at the Missouri River basin by Reager et al. (2014). Reager et al. (2014) indicated that regional TWSA may lead river discharge slightly before the flood season, which can provide useful information on the signal of high streamflow in the coming flood season."

L504: What is 'deficit' here?

Response: We apologize for such confusion because of the wording. As described in Line 502, we have changed "deficit" with "difference" to make the explanation clear.

"The highest difference in the temporally downscaled TWSA and the daily precipitation during the wet season, as revealed by the NDFPI, can indicate the early signs of the region's transition from normal state to a flood-prone situation."

L511: 'activities' instead of 'activates'.

Response: Thanks for your reminder. This sentence has been removed from this manuscript to avoid repetition based on the suggestions of anonymous Referee #2. More detailed information about this change can also be found in the response to anonymous Referee #2.

L513: What kind of surface conditions are meant here?

Response: As described in Line 508 to Line 510, we have revised this sentence to make the explanation clear.

"By using the method of linear detrending, long-term trends in series of TWSA estimates have been removed during the reconstruction of TWSA, because they are generally driven by various human activities such as irrigation, reservoir operation and water withdrawals (Humphrey and Gudmundsson, 2019)."

**References:**

Humphrey, V., Gudmundsson, L., 2019. GRACE-REC: a reconstruction of climate-driven water storage changes over the last century. Earth Syst. Sci. Data 11 (3), 1153-1170.

L533-534: It is not clear what is meant with 'spatially correlated features'.

Response: We apologize for such confusion because of the wording. As described in Line 534 to Line 537, we have revised this sentence to make the explanation clear.

"The latest study has made some initial attempts to learn the spatio-temporal patterns of difference between TWSA derived from GRACE data and those simulated by land surface models based on the convolutional neural network (CNN) models with the goal of providing more accurate TWSA estimates (Mo et al., 2022; Sun et al., 2019)."

**References**

Mo, S., Zhong, Y., Forootan, E., Mehrnegar, N., Yin, X., Wu, J., Feng, W., Shi, X., 2022. Bayesian convolutional neural networks for predicting the terrestrial water storage anomalies during GRACE and GRACE-FO gap. J. Hydrol. 604, 127244.

Sun, A.Y., Scanlon, B.R., Zhang, Z., Walling, D., Bhanja, S.N., Mukherjee, A., Zhong, Z., 2019. Combining physically based modeling and deep learning for fusing GRACE satellite data: can we learn from mismatch? Water Resour. Res. 55 (2), 1179-1195.

L755: What is 'Now'? 2022?

Response: To make it clear, we have changed "Now" with "2020" shown in Table 2 because this table mainly introduces all datasets used in this study.

L760: Is it meaningful to show the overall performance mixing training and validation periods?

Response: Thanks for your reminder. We have deleted the overall performance mixing training and validation periods based on your suggestions.

L768: The Three Gorges Reservoir is difficult to identify on the map.

Response: To make it easy to identify on the map, we have changed the symbol and color of the Three Gorges Reservoir as shown in Figure 1 (denoted as a green hexagon).

[Figure]

980 **Figure 1: Location of the Yangtze River Basin (YRB) in China and its topography. Distribution of meteorological stations and hydrological stations are also shown in this figure. TGR = Three Gorges Reservoir; DEM = Digital Elevation Model.**

---

## Author Comment (AC2)

**To Abhishek Abhi, 24 May 2022**

I read through this interesting manuscript by Xie et al. with great interest. Here, the authors have attempted to resolve the coarse temporal resolution of GRACE(-FO) data by downscaling to the daily time series by employing machine learning methods. Further, a new flood index, namely, the normalized daily flood potential index (NDPFI), is proposed to better characterize the extreme flood events in the Yangtze river basin (YRB).

I have some suggestions that authors may find beneficial while revising their manuscript.

Response: Thanks for your valuable suggestions. We have carefully checked and revised our manuscript in accordance to your comments. The detailed responses to your comments are provided as follows.

Major suggestions

(1) Selection of flood events: I found the authors have selected only extreme flood event(s) to analyze and demonstrate the capability of NDFPI (Lines 395, 413). I think that this may not be the best case. Selecting other flood events that are not well captured in other indices may be a better choice to discern the outperformance of and additional insights by the proposed index over conventional indices, e.g., SPI, SPEI.

Response: Thanks for your valuable suggestions. The reason why we selected only extreme flood event(s) to analyse and demonstrate the capability of NDFPI is that the YRB is particularly prone to catastrophic floods due to its highly uneven rainfall pattern and high annual rainfall (more than 1100 mm) (Zhang et al., 2021; Xiong et al., 2021). In particular, it has been found that both the frequency and severity of extreme flood events generally showed an upward trend in the YRB in the recent decades owing to substantial changes in climate, infrastructure and land use (Huang et al., 2015; Liu et al., 2019; Yang et al., 2021; Zhang et al., 2008). Furthermore, previous studies have compared the difference between the similar indices using monthly GRACE/GRACE-FO satellite data and other traditional evaluation indices, such as SPI and SPEI. For example, Yan et al. (2021) and Yin et al. (2021) clearly indicated that the similar indices using monthly GRACE/GRACE-FO data can better reflect the evolution of flood events than traditional evaluation indices, such as SPI and SPEI, because the GRACE/GRACE-FO observations can measure the vertically integrated water storage over regions. Therefore, we mainly focus on downscaling the TWSA estimates derived from GRACE/GRACE-FO satellite data into daily values and demonstrating its application to monitor the extreme flood events at sub-monthly time scales for the YRB in this study. Nevertheless, we also agree with you that more efforts can be made to analyse the potential of NDFPI to monitor other flood events that are not well captured in other indices when more observations about PET are available in future study.

**References**

Xiong, J., Yin, J., Guo, S., Gu, L., Xiong, F., Li, N., 2021. Integrated flood potential index for flood monitoring in the GRACE era. J. Hydrol. 603, 127115.

Yan, X., Zhang, B., Yao, Y., Yang, Y., Li, J., Ran, Q., 2021. GRACE and land surface models reveal severe drought in eastern China in 2019. J. Hydrol. 601, 126640.

Yin, G., Park, J., 2021. The use of triple collocation approach to merge satellite- and model-based terrestrial water storage for flood potential analysis. J. Hydrol. 603, 127197.

Zhang, X., Zhang, G., Long, X., Zhang, Q., Liu, D., Wu, H., Li, S., 2021. Identifying the drivers of water yield ecosystem service: A case study in the Yangtze River Basin, China. J. Hydrol. 132, 108304.

(2) Terminology of wet and dry seasons: If I am correct, the wet season in the basin spans from June to September, and this is also the same period when the flood of 2020 is observed (e.g., Figure 10). The newly proposed index is also shown better to detect the extreme events during the wet season (e.g., as stated in lines 408, 427). However, in a few places (e.g., lines 433, 551, 817), it is mentioned that the flood events in 2020 as the 'summer 2020'. Is there something that I missed or confused between summer or wet season?

Response: As described in Line 92, the wet season in the YRB spans from April to October, which is consistent with previous studies (Huang et al., 2015; Yang et al., 2010). To avoid ambiguity, we have changed 'summer 2020' with 'Year 2020' based on your suggestions.

**References**

Huang, Y., Salama, M.S., Krol, M.S., Su, Z., Hoekstra, A.Y., Zeng, Y., Zhou, Y., 2015. Estimation of human-induced changes in terrestrial water storage through integration of GRACE satellite detection and hydrological modeling: A case study of the Yangtze River basin. Water Resour. Res. 51 (10), 8494-8516.

Yang, S., Liu, Z., Dai, S., Gao, Z., Zhang, J., Wang, H., Luo, X., Wu, C., Zhang, Z., 2010. Temporal variations in water resources in the yangtze river (Changjiang) over the industrial period based on reconstruction of missing monthly discharges. Water Resour. Res. 46, W10516.

(3) Handling the intermittent data gaps due to battery management (Line 111-112): It is not clear how the intermittent data gaps occurring about every six months in the GRACE and GRACE-FO TWSA time series were filled. Most probably, they were filled by linear interpolation or by the average of the bounding one-two months values. In either case, the filled values are likely (a) underestimating the actual (positive/negative peak) TWSA if the data gap happens to be in the peak of the wet or dry season (there

are a lot of such times, please see footnote of Table 3 in Abhishek et al., 2022), or (b) overestimation or underestimation in case of the high short-term fluctuations in the TWSA. Furthermore, this overestimation or underestimation can be critical given the topical issue of daily monitoring dealt with herein and, subsequently, might lead to inappropriate inferences in the YRB, which is highly vulnerable to floods. In my opinion, these data gaps can either be filled by running the three machine learning models (already used in the manuscript) for monthly TWSA, or alternatively, this can be stated as a likely source of uncertainty.

Response: Thanks for your valuable suggestions. Due to the problem of "battery management", TWSA estimates in some months are not available for the GRACE and GRACE-FO satellites. As you suggested in this comment, these data gaps can be filled by running the three machine learning models (already used in the manuscript) for monthly TWSA. In fact, our previous study (Xie et al., 2019) also indicated that monthly TWSA estimates could be well reconstructed by the MLP model that was also applied in this study. Therefore, we further compared the monthly TWSA estimates reconstructed by the three machine learning models and that acquired from GRACE and GRACE-FO satellites (shown in Figure R1). According to Figure R1, we can find that these data gaps have been well filled by the three machine learning models. Furthermore, as shown in Line 515 to line 519, we have also added some sentences to clearly describe the uncertainty induced by these data gaps based on your suggestions.

[Figure]

Figure R1. Comparison between monthly TWSA estimates reconstructed by the three machine learning models and that acquired from GRACE and GRACE-FO satellites for the YRB.

"Meanwhile, TWSA estimates in some months are not available for the GRACE and GRACE-FO satellites due to the problem of "battery management". Although all these missing months can be effectively filled by different machine-learning based models, it may overestimate or underestimate the actual TWS especially for some extreme values in the peak of the wet or dry season (Abhishek et al., 2022)."

**References**

Abhishek., Kinouchi, T., Abolafia-Rosenzweig, R., Ito, M., 2022. Water Budget Closure in the Upper Chao Phraya River Basin, Thailand Using Multisource Data. Remote Sens. 14(1), 173.

Xie, J., Xu, Y.P., Gao, C., Xuan, W., Bai, Z., 2019. Total basin discharge from GRACE and Water balance method for the Yarlung Tsangpo River basin, Southwestern China. J. Geophys. Res. Atmos. 124, 7617-7632.

Minor suggestions

Line 32: 'severe extreme'. Choosing one of these two words may be better.

Response: As described in Line 58, we have chosen one of these two words based on your suggestions.

"The YRB has been regarded as one of the most sensitive and vulnerable regions that suffered from **severe** floods due to its highly uneven rainfall pattern (Zhang et al., 2021)."

Line 52: 'evaluation' may not be necessary.

Response: As described in Line 43, we have deleted this word based on your suggestions.

"Previous studies have clearly indicated that the proposed indices using GRACE/GRACE-FO data can better reflect the evolution of flood events than traditional indices, such as standardized precipitation index (SPI) and standardized precipitation evapotranspiration index (SPEI), because the GRACE/GRACE-FO observations can measure the vertically integrated water storage over regions (Yan et al., 2021; Yin et al., 2021)."

Line 57: either 'a limited attention has been paid' or 'very few studies have paid attention to'.

Response: As described in Line 49, we have rewritten this sentence based on your suggestions.

> "To date, **very few studies have paid attention to** monitor flood events at sub-monthly time scales using GRACE data."

Line 59: Since downscaling of the TWSA data is the aim (Line 60), the phrase 'using the temporally downscaled GRACE data' may be removed.

Response: As described in Line 50, we have removed the phrase 'using the temporally downscaled GRACE data' from this sentence based on your suggestions.

> "Given the rapid occurrence and evolution of some extreme events within a short period, there is a great need to monitor the flood events at a finer temporal resolution (e.g. day), which has important implications for better understanding the mechanisms of extreme flood events development in the YRB."

Line 81: Reference of Fig. 1 may be provided here.

Response: We have added the reference of Fig. 1 in this sentence as described in Line 82.

> "The terrain of the YRB generally decreases from west to east with altitudes ranging from -142 m to 7143 m above the sea level **(shown in Fig. 1).**"

Line 84: 'temperature' to 'temperate'

Response: As described in Line 87, we have changed 'temperature' to 'temperate' based on your suggestions.

> "The YRB is located in typically subtropical and **temperate** climate zones, which is dominated by three types of monsoons, namely the Siberian northwest monsoon winds in winter, the Indian southeasterly monsoon winds and the East Asian monsoon in summer (Kong et al., 2020)."

Lines 106, 119, 131: 'basins' to 'subbasins'

Response: As described in Line 109, 125 and 137, we have changed 'basins' to 'subbasins' based on your suggestions.

Line 131: How the 'extreme flood events' were extracted is not clear? Providing a reference to Section 4.4 may be better.

Response: As described in Line 137 to Line 139, we have added a reference in this sentence, which can help us better understand how the extreme flood events are extracted in this study. In addition, we describe in detail the key steps about how the extreme flood events were extracted in Section 4.4.

"Meanwhile, extreme flood events in the YRB and its individual subbasins during the study period can be extracted from daily time series of streamflow observed from the above hydrological stations **(Tarasova et al., 2018)**. More details about how the extreme flood events are extracted will be described in the following Section 4.4.

"**4.4 Flood event selection**

A nonparametric algorithm suggested by Tarasova et al. (2018) is adopted to identify runoff events in this study, which has been widely applied in many different basins over the world because of its advantages in identifying flood events (Fischer et al., 2021; Giani et al., 2022; Lu et al., 2020; Winter et al., 2022). A brief procedure of this algorithm is described as follows: (1) picking out local minima within nonoverlapping five-day windows with respect to the entire streamflow time series; (2) examining the extracted series of minima with the goal of finding turning points, all of which are usually defined as the points that are at least 1.11 times smaller than their neighboring minima; (3) reconstructing the base flow hydrograph according to the linear interpolation between the turning points, which are previously obtained in Step (2); (4) screening the streamflow time series to identify runoff events after the separation of base flow. Traditionally, a typical runoff event can be characterized by three main components, namely peak, beginning, and end points. A peak refers to the maximum of streamflow for a specific period. The beginning point refers to the closest point in time when total runoff is equal to base flow before the peak. Similarly, the end point denotes the closest point in time when total runoff is equal to base flow after the peak."

Line 142: 'o' to 'to'

Response: According to the comments from Reviewer #1, this word has been removed from this manuscript. More details about this change also can refer to the response to the Comment #6 from Reviewer #1

Line 146, 530: using the single model output for soil moisture may include the implicit biases. How about using the ensemble mean from multiple model outputs, maybe even within the GLDAS series, subject to the availability and consistency with the study period.

Response: Thanks for your valuable suggestions. As documented in many studies (Proulx et al., 2013; Wei et al., 2022; Xiong et al., 2021), four different land surface models, i.e., Mosaic, Noah, Community Land Model (CLM), and Variable Infiltration Capacity (VIC) Model are provided by Global Land Data Assimilation System (GLDAS) datasets (https://disc.gsfc.nasa.gov/gldas). They have been proven to have

good accuracy and can be used for the comparison of GRACE data. Previous studies (Liu et al., 2022; Long et al., 2014; Wang et al., 2021; Zhang et al., 2022) indicate that soil moisture in the Noah land surface model has a higher correlation with the GRACE-derived TWSA estimates compared to other land surface models including the Yangtze River Basin. Furthermore, our previous study (Xie et al., 2019) also 170 suggests that the Global Land Data Assimilation System version 2.1 (GLDAS 2.1) Noah land surface model shows a superiority than the other models. Given the above reasons, we used the single model output for soil moisture in this study. Nevertheless, we also agree with you that the ensemble mean from multiple model outputs can be applied in our next study to further consider the uncertainties or implicit biases induced by model outputs.

175 **References**

Liu, M., Pei, H., Shen, Y., 2022. Evaluating dynamics of GRACE groundwater and its drought potential in Taihang Mountain Region, China. J. Hydrol. *in press*, https://doi.org/10.1016/j.jhydrol.2022.128156.

Long, D., Shen, Y., Sun, A., Hong, Y., Longuevergne, L., Yang, Y., Li, B., Chen, L., 2014. Drought and 180 flood monitoring for a large karst plateau in Southwest China using extended GRACE data. Remote Sens. Environ. 155, 145-160.

Proulx, R.A., Knudson, M.D., Kirilenko, A., VanLooy, J.A., Zhang, X., 2013. Significance of surface water in the terrestrial water budget: A casestudy in the Prairie Coteau using GRACE, GLDAS, Landsat,and groundwater well data.

185 Wang, L., Peng, Z., Ma, X., Zheng, Y., Chen, C., 2021. Multiscale gravity measurements to characterize 2020 flood events and their spatio-temporal evolution in Yangtze river of China. J. Hydrol. 603, 127176.

Wei, M., Zhou, H., Luo, Z., Dai, M., 2022. Tracking inter-annual terrestrial water storage variations over Lake Baikal basin from GRACE and GRACE Follow-On missions. J. Hydrol. Reg. Stud. 40, 101004.

190 Xiong, J., Yin, J., Guo, S., Slater, L., 2021. Continuity of terrestrial water storage variability and trends across mainland China monitored by the GRACE and GRACE-Follow on satellites. J. Hydrol. 599, 126308.

Xie, J., Xu, Y.P., Gao, C., Xuan, W., Bai, Z., 2019. Total basin discharge from GRACE and Water balance method for the Yarlung Tsangpo River basin, Southwestern China. J. Geophys. Res. Atmos. 124, 195 7617-7632.

Zhang, X., Li, J., Dong, Q., Wang, Z., Zhang, H., Liu, X., 2022. Bridging the gap between GRACE and GRACE-FO using a hydrological model. Sci. Total Environ. 822, 153659. Water Resour. Res. 49, 5756-5764.

Line 157: 'machine' learning-based

Response: As described in Line 158, we have changed this sentence as you suggested.

"In Step 2, the relationship between TWSA estimates and all hydro-climatic factors at monthly time scales for the YRB can be built by using three different **machine** learning-based models, namely MLP model, LSTM model and MLR model respectively."

Line 168: 'served' to 'used' or 'employed' or some other appropriate verb.

Response: As described in Line 171, we have changed this sentence as you suggested.

"Specifically, three types of models, namely, the artificial neural network (ANN), the recurrent neural network (RNN) and the multiple linear regression (MLR) are **used** as the statistical downscaling methods."

Line 411: 'rivers' may not be the best word. Please see another suitable word, if possible.

Response: As described in Line 409, we have rewritten this sentence as you suggested.

"According to the Yangtze River Conservancy Commission of Ministry of Water Resources, the YRB has suffered from catastrophic flooding in 2020."

**Reference**

Abhishek, Kinouchi, T., Abolafia-Rosenzweig, R., Ito, M., 2022. Water Budget Closure in the Upper Chao Phraya River Basin, Thailand Using Multisource Data. Remote Sens. 14.

Response: The results shown in the study of Abhishek et al. (2022) are very useful, which has been cited in our revised manuscript, as described in Line 517.

---

## Author Comment (AC3)

**To anonymous Referee #2, 01 Jun 2022**

General comments for the authors' reference:

(1) Line 32-34: "During the past decades, the increasingly intensified human activities and climate change have significantly changed the hydrological cycles in the YRB and thus accelerated the variation of flood characteristics in this region". This statement seems too general for readers. Please provide relevant citations to support this statement.

Response: As described in Line 59 to Line 62, we have provided relevant citations to support this statement based on your suggestion.

"During the past decades, the increasingly intensified human activities and climate change have significantly changed the hydrological cycles in the YRB and thus accelerated the variation of flood characteristics in this region (Fang et al., 2012; Wang et al., 2011)."

**References**

Fang, H., Han, D., He, G., Chen, M., 2012. Flood management selections for the Yangtze River midstream after the Three Gorges Project operation. J. Hydrol. 432-433, 1-11.

Wang, M., Zheng, H., Xie, X., Fan, D., Yang, S., Zhao, Q., Wang, K., 2011. A 600-year flood history in the Yangtze River drainage: comparison between a subaqueous delta and historical records. Chin. Sci. Bull. 58 (2), 188-195.

(2) Section 3.1: Please consider adding some additional sentences to describe the difference between these three GRACE solutions used in this study, which are applied to characterize the variations of TWSA in the YRB and its individual basins during the study period.

Response: As described in Line 118 to Line 120, we have added additional sentences to describe the difference between these three GRACE solutions used in this study based on your suggestion.

"As documented in previous studies (Long et al., 2014; Xie et al., 2022), there are slight differences between these three GRACE and GRACE-FO solutions when estimating the variation of regional TWSA. The difference between these three GRACE and GRACE-FO solutions mainly arises from the processing algorithms or constrained solutions."

**References**

Long, D., Shen, Y., Sun, A., Hong, Y., Longuevergne, L., Yang, Y., Li, B., Chen, L., 2014. Drought and flood monitoring for a large karst plateau in Southwest China using extended GRACE data. Remote Sens. Environ. 155, 145-160.

Xie, J., Xu, Y.P., Guo, Y., Wang, Y., Chen, H., 2022. Understanding the impact of climatic variability on terrestrial water storage in the Qinghai-Tibet Plateau of China. Hydrolog. Sci. J. 67(6), 1-16.

(3) Line 390: When explaining the results shown in Figure 6(a), the authors simply attributed it to the relatively poor relationship between TWSA estimates and hydro-climatic factors for this region that is described in Fig. 5(a). This statement is not convincing enough. If possible, please consider to provide more details to explain the results shown in Figure 6(a).

Response: Thanks for your suggestions. As described in Line 381 to Line 391, we have added more details to explain the results shown in Figure 6(a).

"As documented in previous studies (Liu et al., 2020; Shi et al., 2020), it has long been challenging to accurately perform hydrological simulation across the SYRB because of the complex hydrological processes for this alpine basin. For example, parameter settings calibrated by GLDAS Noah land surface model might not be highly accurate for SMS simulation across the SYRB, because field measurements of SMS in this region are extremely limited. Harsh climatic conditions and limited weather stations can additionally influence the accuracy of meteorological observations such as precipitation and temperature across the SYRB especially for some extreme values. Given the above reasons, there is a relatively poor relationship between TWSA estimates and hydro-climatic factors across the SYRB based on the MLP (shown in Fig. 5(a)). Furthermore, the uncertainty in the observed precipitation and temperature and SMS derived from the GLDAS Noah land surface model can eventually result in some discrepancies between temporally downscaled TWSA at sub-monthly time scales and monthly TWSA estimates derived from GRACE/GRACE-FO satellites data, as described in Fig. 6(a)."

**References**

Liu, L., Jiang, L., Wang, H., Ding, X., Xu, H., 2020. Estimation of glacier mass loss and its contribution to river runoff in the source region of the Yangtze River during 2000-2018. J. Hydrol. 589, 125207.

Shi, R., Yang, H., Yang, D., 2020. Spatiotemporal variations in frozen ground and their impacts on hydrological components in the source region of the Yangtze River. J. Hydrol. 590, 125237.

(4) The section of Conclusion could have been improved by mentioning more informative results. For example, as stated in Section 5.5, the proposed NDFPI can reach the threshold values earlier than that of daily streamflow observations for the 90th, 95th and 99th percentile floods, which has not been mentioned in the section of Conclusion.

60 Response: As described in Line 569 to Line 573, we have added additional sentences to describe the findings shown in this study based on your suggestions.

"(3) By jointly using daily average precipitation anomalies and temporally downscaled TWSA, the proposed NDFPI can effectively detect the flood events at sub-monthly time scales occurred in 2020 for the entire YRB;

65 (4) The comparison analysis indicates that different types of flood events including the 90th, 95th and 99th percentile floods can be monitored by the proposed NDFPI earlier than traditional streamflow observations with respect to the YRB and its individual subbasins, which is very vital for flood forecasts and warning across this region."

Minor comments:

70 (1) Line 29-30: There is no need for citing so many papers only to explain the topic of monitoring extreme flood events. Please keep the most relevant ones.

Response: As described in Line 30, we have deleted the unnecessary references shown in this sentence and only keep the most relevant ones based on your suggestions.

"Therefore, monitoring extreme flood events has long been a hot topic for hydrologists and decision
75 makers around the world (Tanoue et al., 2020; Tellman et al., 2021)."

**References**

Tanoue, M., Taguchi, R., Nakata, S., Watanabe, S., Fujimori, S., Hirabayashi, Y., 2020. Estimation of direct and indirect economic losses caused by a flood with long-lasting inundation: Application to the 2011 Thailand flood. Water Resour. Res. 56, e2019WR026092.

80 Tellman, B., Sullivan, J.A., Kuhn, C., Kettner, A.J., Doyle, C.S., Brakenridge, G.R., Erickson, T.A., Slayback, D.A., 2021. Satellite imaging reveals increased proportion of population exposed to floods. Nature 596, 80-86.

(2) Line 171-173: This sentence seems a bit confused. Please rewritten it to avoid ambiguity.

Response: As described in Line 174 to Line 177, we have rewritten this sentence to avoid ambiguity.

85    "As pointed out by previous studies (Humphrey and Gudmundsson, 2019; Khorrami et al., 2021; Shah et al., 2021), long-term changes in TWSA are primarily caused by frequent human activities such as persistent groundwater overexploitation and massive construction of large reservoirs."

(3) Line 312: Remove the word of "different" in this sentence.

Response: Based on your suggestions, we have deleted this word from the sentence as described in Line
90 298.

    "This phenomenon can be explained by the discrepancies resulted from the components of SMSA and TWSA."

(4) Line 347: Please replace the "shows" by "show".

Response: Based on your suggestions, we have replaced the "shows" by "show" as described in Line 333.

95    "Taking the entire YRB as an example (Fig. 5(j-l)), GRACE/GRACE-FO derived TWSA estimates show a RMSE of 10.9 mm/month for the MLP-derived TWSA estimates, which is lower than that of 15.1 mm/month for the LSTM-derived TWSA estimates (~39% difference) and that of 13.3 mm/month for the MLR-derived TWSA estimates (~22% difference)."

(5) Line 408: This sentence is confused. Please clarify.

100 Response: As described in Line 408 to Line 410, we have rewritten this sentence in order to clarify it.

    "To better monitor severe flood events over the YRB, we propose a new index, i.e. NDFPI, by jointly using the temporally downscaled TWSA data and daily precipitation data as introduced in Section 4.5."

(6) Line 510: This sentence can be removed since it repeats information from the section of Method.

Response: Based on your suggestions, we have deleted this sentence to avoid repetition.

105 (7) Figure 9 and Figure 10: Please add the meaning of acronyms, such as NDFPI in the captions of these figures.

Response: As described in Figure 9 and Figure 10, we have added the meaning of these acronyms in the captions of these figures based on your suggestions.

[Figure]

110    **Figure 9: Percentile duration curves of daily streamflow observations and NDFPI for the 90th percentile floods across (a) the SYRB, (b) the UYRB, (c) the UMYRB and (d) the YRB respectively during 2003-2020. The red dots and blue dots represent threshold values of daily streamflow and NDFPI for the 90th percentile floods across different regions. SYRB = Source regions of Yangtze River Basin; UYRB = Upper regions of Yangtze River Basin; UMYRB = Upper and middle regions of Yangtze River Basin; YRB = Yangtze River Basin; NDFPI = Normalized daily flood potential index.**

[Figure]

**Figure 10: Comparison between basin averaged NDFPI and daily streamflow observations for the 90th percentile floods in 2020 across (a) the SYRM (observed at Shigu station), (b) the UYRB (observed at Yichang station), (c) the UMYRB (observed at Hankou station) and (d) the YRB (observed at Datong station). Pink rectangles denote the duration period between the thresholds of daily streamflow for the 90th percentile floods and peak streamflow observed at the controlling hydrological stations over different regions. The thresholds of daily streamflow and NDFPI for the 90th percentile floods are represented by the red dash lines and blue dash lines respectively. Note that the scales of streamflow shown in each figure are not always same. SYRB = Source regions of Yangtze River Basin; UYRB = Upper regions of Yangtze River Basin; UMYRB = Upper and middle regions of Yangtze River Basin; YRB = Yangtze River Basin; NDFPI = Normalized daily flood potential index.**

(8) Table 2: Please carefully check the abbreviations of all items shown in this table.

Response: Thanks for your valuable suggestions. As described in Table 2, we have carefully checked the abbreviations of all items shown in this table and made the corresponding corrections.

**Table 2: An overview of all datasets used in this study.**

| Data | Source | Temporal resolution | Spatial resolution | Time span |
|---|---|---|---|---|
| Terrestrial water storage anomaly (TWSA) | GRACE/GRACE-FO CSR | Month | 0.5° | 2002 - 2022 |
| | GRACE/GRACE-FO JPL | Month | 0.5° | 2002 - 2022 |
| | GRACE/GRACE-FO GSFC | Month | 0.5° | 2002 - 2022 |
| Soil moisture storage (SMS) | GLDAS 2.1 - Noah | 3 hours | 1° | 2002 - 2022 |
| Precipitation (P) | CMA | Day | / | 2003 - 2020 |
| Temperature (T) | CMA | Day | / | 2003 - 2020 |
| Streamflow | In situ | Day | / | 2003 - 2020 |

130 *Note.* GRACE = Gravity Recovery and Climate Experiment mission; GRACE-FO = Gravity Recovery and Climate Experiment Follow-On mission; CSR = Center for Space Research; JPL = Jet Propulsion Laboratory; GSFC = Goddard Space Flight Center; GLDAS = Global Land Data Assimilation system; CMA = China Meteorological Administration

---

## Author Response (AR2)

We would like to thank both editor and reviewers for their constructive suggestions on how to improve the manuscript. In the following, we provide answers to comments from Referees below.

**To anonymous Referee #2, 22 Oct 2022**

General comments for the authors' reference:

5   I note that this is a revision manuscript. The authors deal with the reconstruction of GRACE/GRACE-FO satellite data, evaluation of the terrestrial water storage anomaly and then apply to monitor the sub-month flood events in the Yangtze River Basin. The manuscript is well written and the results concluded from this manuscript are promising and helpful. Meanwhile, the authors' response is comprehensive and logical with large supported references. The manuscript has been improved largely through the revision. Thus, I
10  think it is ready for publication. I only have several minor suggestions:

(1) Line 34 - Line 36: Please add more references to better show the meaning of this sentence.

Response: As described in Line 34 - Line 36, we have added more references to better show the meaning of this sentence.

"TWSA derived from GRACE/GRACE-FO satellites comprises all the surface and subsurface water
15  over land, which can be used to monitor the hydrologic variations in response to extreme weather events (Li et al., 2022; Xie et al., 2019a)."

**References**

Li, X., Scanlon, B.R., Mann, M.E., Li, X., Tian, F., Sun, Z., Wang, G., 2022. Climate change threatens terrestrial water storage over the Tibetan Plateau. Nat. Clim. Change. 12, 801-807.

20  Xie, J., Xu, Y.P., Wang, Y., Gu, H., Wang, F., Pan, S., 2019a. Influences of climatic variability and human activities on terrestrial water storage variations across the Yellow River basin in the recent decade. J. Hydrol. 579, 124218.

(2) Line 63: an upward trend >> upward trends

Response: Based on your suggestions, we have replaced "an upward trend" with "upward trends" as
25  described in Line 63.

"It has been found that both the frequency and severity of extreme flood events generally showed **upward trends** in the YRB in recent decades owing to substantial changes in climate, infrastructure and land use (Huang et al., 2015; Liu et al., 2019; Yang et al., 2021; Zhang et al., 2008)."

(3) Line 64: has experienced >> experienced.

Response: Based on your suggestions, we have replaced "has experienced" with "experienced" as described in Line 64.

"For example, in Year 2020, the YRB **experienced** one of the most extreme flood events on record."

(4) Line 388: uncertainty >> uncertainties

Response: Based on your suggestions, we have replaced "uncertainty" with "uncertainties" as described in Line 388.

"Furthermore, the **uncertainties** in the observed precipitation and temperature and SMS derived from the GLDAS Noah land surface model can eventually result in some discrepancies between temporally downscaled TWSA at sub-monthly time scales and monthly TWSA estimates derived from GRACE/GRACE-FO satellite data, as described in Fig. 6(a)."

(5) Line 476 - Line 477: "The traditional flood monitoring approaches mainly provide useful information about the evolution of flood events over the study region through the measurements of rainfall and streamflow." This statement is not appropriate and please rewrite this sentence.

Response: Thanks for your valuable suggestions. As described from Line 476 to Line 477, we have carefully checked and rewritten this sentence to make it clear.

"The traditional flood monitoring approaches can provide useful information about the evolution of flood events over the study region through the measurements of rainfall and streamflow."

(6) Line 525 - Line 526: Please check the tense of "involve" shown in this sentence.

Response: Thanks for your valuable suggestion. As described from Line 525 to Line 526, we have carefully checked and changed the tense of "involve" shown in this sentence.

"However, as pointed out by previous studies (Landerer et al., 2012; Save et al., 2016; Scanlon et al., 2016), gridded TWSA estimates derived from GRACE/GRACE-FO satellite data **involved** relatively large uncertainty induced by associated measurement errors and signal leakage errors."

(7) Line 543 - Line 544: Please rewrite this sentence to make it clear.

Response: Thanks for your valuable suggestion. As described from Line 543 to Line 544, we have carefully checked and rewritten this sentence to make it clear.

"Overall, the present study shows the great potential of temporally downscaled GRACE/GRACE-FO satellite data in monitoring the extreme flood events."

(8) The labels of hydrological stations (Yichang and Hankou) in Figure1 are indistinct, please change the color to make them clearly.

Response: As depicted in Figure 1, we have changed the colors of these labels to make them clear.

[Figure]

**Figure 1: Location of the Yangtze River Basin (YRB) in China and its topography. Distribution of meteorological stations and hydrological stations are also shown in this figure. TGR = Three Gorges Reservoir; DEM = Digital Elevation Model.**